# Bright split red fluorescent proteins for the visualization of endogenous proteins and synapses

Siyu Feng [1], Aruna Varshney[2], Doris Coto Villa[2], Cyrus Modavi[3], John Kohler[4], Fatima Farah[2], Shuqin Zhou[5,6], Nebat Ali[2], Joachim D. Müller[4], Miri K. Van Hoven[2] & Bo Huang [6,7,8]

Self-associating split fluorescent proteins (FPs) are split FPs whose two fragments spontaneously associate to form a functional FP. They have been widely used for labeling proteins, scaffolding protein assembly and detecting cell-cell contacts. Recently developments have expanded the palette of self-associating split FPs beyond the original split GFP$_{1-10/11}$. However, these new ones have suffered from suboptimal fluorescence signal after complementation. Here, by investigating the complementation process, we have demonstrated two approaches to improve split FPs: assistance through SpyTag/SpyCatcher interaction and directed evolution. The latter has yielded two split sfCherry3 variants with substantially enhanced overall brightness, facilitating the tagging of endogenous proteins by gene editing. Based on sfCherry3, we have further developed a new red-colored trans-synaptic marker called Neuroligin-1 sfCherry3 Linker Across Synaptic Partners (NLG-1 CLASP) for multiplexed visualization of neuronal synapses in living *C. elegans*, demonstrating its broad applications.

[1] The UC Berkeley-UCSF Graduate Program in Bioengineering, San Francisco, CA 94143, USA. [2] Department of Biological Sciences, San Jose State University, San Jose, CA 95192, USA. [3] Department of Bioengineering and Therapeutic Sciences, University of California in San Francisco, San Francisco, CA 94143, USA. [4] School of Physics and Astronomy, University of Minnesota, Minneapolis, MN 55455, USA. [5] School of Pharmaceutical Sciences, Tsinghua University, Beijing 100084, China. [6] Department of Pharmaceutical Chemistry, University of California in San Francisco, San Francisco, CA 94143, USA. [7] Department Biochemistry and Biophysics, University of California, San Francisco, San Francisco, CA 94143, USA. [8] Chan Zuckerberg Biohub, San Francisco, CA 94158, USA. Correspondence and requests for materials should be addressed to B.H. (email: bo.huang@ucsf.edu)

Self-associating split fluorescent proteins (SAsFPs) are a powerful tool for protein labeling and live-cell imaging. In this system, the eleventh β-strand of FP ($FP_{11}$, 16 amino acids) is separated out from the remainder of FP ($FP_{1-10}$) and is genetically fused to the protein of interest (POI). Specific fluorescence signal is detected when $FP_{1-10}$ reconstitutes with the on-target $FP_{11}$ to generate a functional fluorescent protein. Since the initial development of self-associating split $GFP_{1-10/11}$[1], this technology has been modified and adapted for an extensive range of applications including protein labeling and visualization[2–4], scaffolding protein assembly[3], protein solubility and aggregation assays[5,6], and monitoring the membrane fusion process[7]. One prominent application is generating a library of human cells with fluorescently tagged endogenous proteins via CRISPR/Cas9-mediated homology-directed repair[4]. The small size of the $GFP_{11}$ tag markedly improves the knock-in efficiency and simplifies the donor DNA preparation. In another application, the split $GFP_{1-10/11}$ system has also been utilized to visualize synapses in living nervous systems by Neuroligin-1 GFP Reconstitution Across Synaptic Partners (NLG-1 GRASP)[8].

While all these applications have focused on a single, green-colored channel, expanding the color palette will greatly benefit the investigation of more complex biological systems by enabling multicolor imaging. For this purpose, we have recently developed a red-colored split $sfCherry_{1-10/11}$[3] and then a brighter split $sfCherry2_{1-10/11}$[9], enabling dual-color endogenous labeling in human cells using orthogonal $FP_{11}$ tags[9] and visualization of *Listeria* protein secretion in infection[10]. However, unlike split $GFP_{1-10/11}$, which is as bright as its full-length, split $sfCherry2_{1-10/11}$ produces substantially lower overall fluorescence signal than its full-length counterpart.

Here, we have characterized the complementation mechanism of split FP systems by examining their overall and single-molecule fluorescence brightness. The results suggest a two-step complementation model in which the affinity between the $FP_{1-10}$ and $FP_{11}$ fragment is the major limitation to the overall fluorescence signal. Based on this model, we have devised a SpyTag/SpyCatcher-assisted approach to improve the complementation efficiency of $sfCherry2_{1-10/11}$. Furthermore, we have engineered two split sfCherry3 variants with much-enhanced complementation efficiency through a combination of cycles of directed evolution and structure-based site-directed mutagenesis. For tagging endogenous proteins by gene editing, sfCherry3 improves the sorting efficiency for successfully knocked-in cells by 5–10-fold in six tested targets, as compared to sfCherry2. Moreover, we have also developed a new red-colored trans-synaptic marker called Neuroligin-1 sfCherry3 Linker Across Synaptic Partners (NLG-1 CLASP). We established that like NLG-1 GRASP, NLG-1 CLASP labels connections between correct synaptic partners and has a spatial pattern similar to that predicted by electron micrograph reconstruction[11,12]. As a validation, NLG-1 CLASP labeling is disrupted by loss of the *clr-1* gene, which is required for synaptic partner recognition[13]. To demonstrate the utility of this new biomarker, we utilize it in combination with the NLG-1 GRASP marker to differentially label subsets of synapses made between the same interneurons and different neuronal partners. We also demonstrate that in this system the fluorescence of both biomarkers are severely reduced in *clr-1* mutants, as *clr-1* is required for both sets of synapses[13]. These experiments indicate that background fluorescence is minimal, and that these markers correctly visualize synaptic connections.

## Results

### The complementation process of SAsFPs.
Previously, we have shown that split $GFP_{1-10/11}$ has nearly identical overall brightness as its full-length counterpart, whereas both split $mNeonGreen2_{1-10/11}$ ($mNG2_{1-10/11}$) and split $sfCherry2_{1-10/11}$ are substantially dimmer[9]. This suboptimal performance of $mNG2_{1-10/11}$ and $sfCherry2_{1-10/11}$ could be attributed to either (a) the lower molecular brightness of complemented split FPs or (b) incomplete complementation between the $FP_{1-10}$ and $FP_{11}$ fragments. To test (a), we measured the single-molecule brightness of these three split FPs and their full-length counterparts in living cells using fluorescence fluctuation spectroscopy[14]. We observed no significant difference in single-molecule brightness between the split and the full-length FPs in all three cases (Fig. 1a). Therefore, incomplete complementation should be the cause of the reduced overall fluorescence signal. The ratio of overall fluorescence between split and full-length FPs then reflects the complementation efficiency.

For $mNG2_{1-10/11}$ and $sfCherry2_{1-10/11}$, to determine their complementation efficiency and compare them to $GFP_{1-10/11}$, we took a similar approach as previously done[9]. We transiently expressed in HEK 293T cells the full-length FP or the two fragments: $FP_{1-10}$ and $FP_{11}$ on a well-folded carrier protein. We quantified whole-cell fluorescence by flow cytometry while using a co-expressed infrared fluorescent protein mIFP to measure the expression level. mIFP was linked to the full-length FP or the $FP_{11}$ fragment through a P2A self-cleavage site to ensure equimolar expression. Fluorescence intensities in infrared and green/red channels of each single cell events were displayed in log–log-scale scatter plots (Fig. 1b–d).

In the cases of full-length FPs, as expected, single cell fluorescence intensities followed the trend line with a slope of 1 because mIFP was expressed equimolarly (Fig. 1e). In the split cases, split $GFP_{1-10/11}$ almost completely followed the same diagonal line as its full-length counterpart (except for a subtle deviation at the lower-expression end) (Fig. 1b), suggesting a complementation efficiency of almost 100% across a wide range of expression levels. In contrast, both split $mNG2_{1-10/11}$ and split $sfCherry2_{1-10/11}$ deviated from the trend lines of their full-length counterparts (Fig. 1c, d). Considering that the complementation process is ultimately irreversible, as we have shown by both fluorescence recovery after photobleaching (FRAP) and photoactivation measurements (Supplementary Figs. 1 and 2), this observation prompted us to consider a two-step complementation process for self-associating split FPs: the two fragments undergo a reversible association/dissociation equilibrium before entering an irreversible process of folding and/or chromophore maturation:

$$FP_{1-10} + FP_{11} \rightleftharpoons FP^*_{1-10/11} \xrightarrow{\text{folding/maturation}} FP_{1-10/11} \rightarrow \text{degradation.}$$

We have verified that in our co-transfection scheme, the expression levels (concentrations) of $FP_{1-10}$ and $FP_{11}$ fragments are proportional through an independent experiment in cells co-expressing mIFP-P2A-$sfCherry2_{11}$-Carrier and TagBFP-P2A-$sfCherry2_{1-10}$ (Fig. 1e). Then, at a steady state, this two-step model predicts the following relationship between the two channels of flow cytometry (see Supplementary Notes):

$$[\text{mIFP}] \propto \sqrt{K'_{D}[FP_{1-10/11}]} + [FP_{1-10/11}],$$

where [mIFP] represents the sum of concentrations of uncomplemented and complemented $FP_{11}$, $[FP_{1-10/11}]$ is the concentration of matured $FP_{1-10/11}$, and $K'_{D}$ is the effective dissociation constant of the overall complementation/maturation process. When $K'_{D}$ is much lower than the expression level of $FP_{11}$ (the case of $GFP_{1-10/11}$), the second term dominates, leading to a near proportional relationship between the two flow cytometry channels (slope of 1 in log–log plot, Fig. 1b). In the opposite case of high $K'_{D}$ (the cases of $mNG2_{1-10/11}$ and $sfCherry2_{1-10/11}$), the first term dominates, resulting in log–log scatter plots

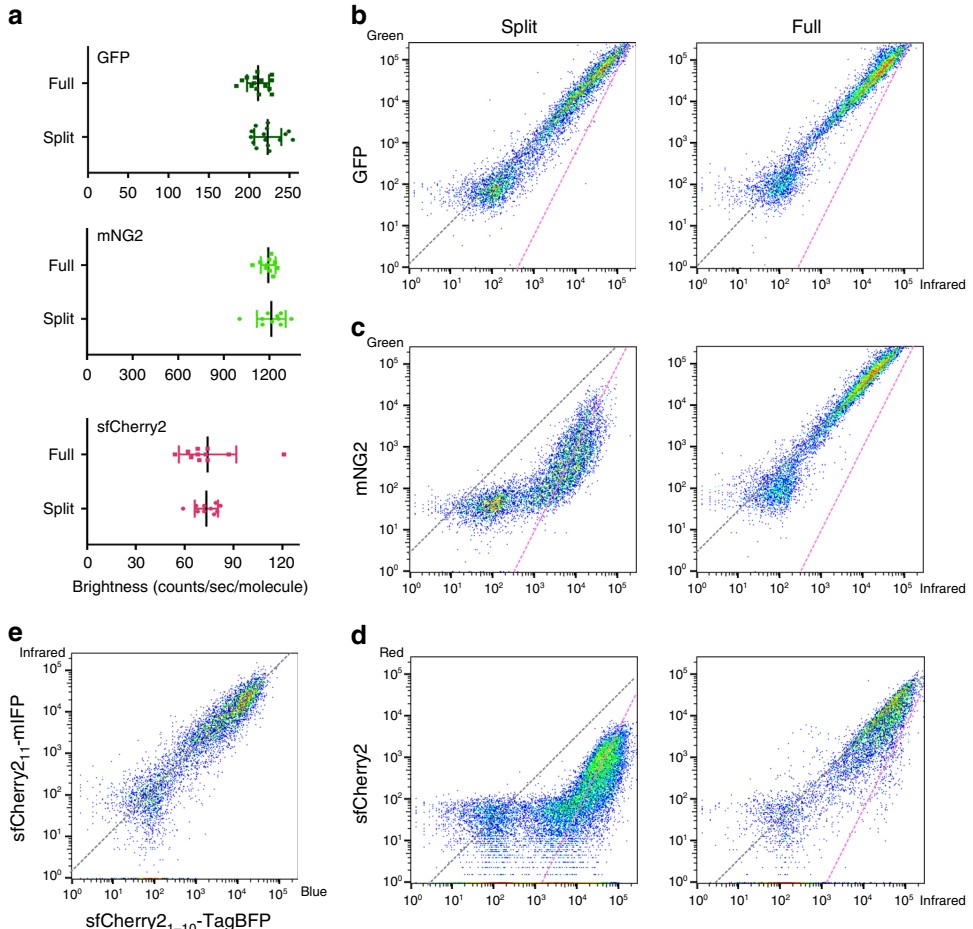

**Fig. 1** Characterization of split fluorescent proteins. **a** Single-molecule brightness measurement of split FPs and their full-length counterparts using fluorescence fluctuation spectroscopy. $N = 10$ (mNG2 and sfCherry5) or 15 (GFP) measurements. Error bars are standard deviations. See Supplementary Data for the list of data values. **b–d** Flow cytometry analysis of whole-cell fluorescence in HEK 293T expressing either **b** GFP$_{1-10/11}$, **c** mNG2$_{1-10/11}$, and **d** sfCherry2$_{1-10/11}$ or their full-length counterparts. The x-axis is the log-scale infrared fluorescence intensity indicating the expression level, and the y-axis is the log-scale green (or red) fluorescence intensity. The gray dashed trend lines have a slope of 1 and intercepts are set to best follow the points in the right (full-length) panels. The pink dashed trend lines have a slope of 2 and intercepts set to best follow the points in the left (split) panels with the exception of GFP$_{1-10/11}$. **e** Expression levels of two fragments are proportional within a wide range of expression levels in a co-transfect experiment. The gray dashed trend line has a slope of 1

following more closely to slope-2 trend lines, matching our observations (Fig. 1c, d).

**SpyTag/SpyCatcher-assisted complementation of split sfCherry2.** Our model indicates that the complementation efficiency of split mNG2$_{1-10/11}$ and sfCherry2$_{1-10/11}$ improves with raised local concentration of fragments, leading to enhanced overall fluorescence signal. Therefore, we sought to utilize a pair of high-affinity binding partners to bring the two fragments into spatial proximity. Because a major advantage of the split FP$_{1-10/11}$ is to label endogenous proteins through knocking-in the short FP$_{11}$ peptide, it is preferable to have a small binding partner for the FP$_{11}$ fragment. For this purpose, we chose the SpyTag/Spy-Catcher[15] system, a peptide–protein pair that undergoes irreversible binding through formation of an isopeptide bond. The 13-amino-acid (aa) SpyTag is sufficiently short that even when concatenated with GFP$_{11}$, the resulting sequence remains small enough for knock-in using synthetic oligo donor DNAs[16].

We examined SpyTag/SpyCatcher-assisted complementation on sfCherry2$_{1-10/11}$ using the recently improved Spy002 pair[17] (Fig. 2a). We fused SpyCatcher to the N-terminus of sfCherry2$_{1-10}$ (the C-terminus is the split site) through a flexible linker in either 6 aa or 15 aa length (Fig. 2a). We generated concatenated tags with SpyTag on either the N- or C-terminus of sfCherry2$_{11}$ with a double-glycine linker. We performed similar flow cytometry experiments as described earlier except that mIFP was replaced by TagBFP. Among the four possible combinations of binders/tags (Supplementary Fig. 2), SpyCatcher-6aa-sfCherry2$_{1-10}$ with SpyTag-sfCherry2$_{11}$-TagBFP demonstrated the most pronounced shift towards a trend line with a slope of 1 in the scatter plot (Fig. 2b). We have further verified that a shorter SpyTag-sfCherry2$_{11}$ fusion without the double-gylcine spacer behaved as efficiently (data not shown).

We validated the improvement in overall brightness for cellular microscopy by labeling the N-terminus of histone 2B (H2B) with SpyTag-sfCherry2$_{11}$-TagBFP and co-expressing it with either sfCherry2$_{1-10}$ or SpyCatcher-6aa-sfCherry2$_{1-10}$ fusion in HEK 293T cells. We observed the Spy-assisted system (Fig. 2c) could mark the nuclei with much stronger fluorescence signal with the same expression vectors. We further demonstrated that the sfCherry2$_{11}$-SpyTag can also be fused the C-terminus of the protein of interest (Supplementary Fig. 4).

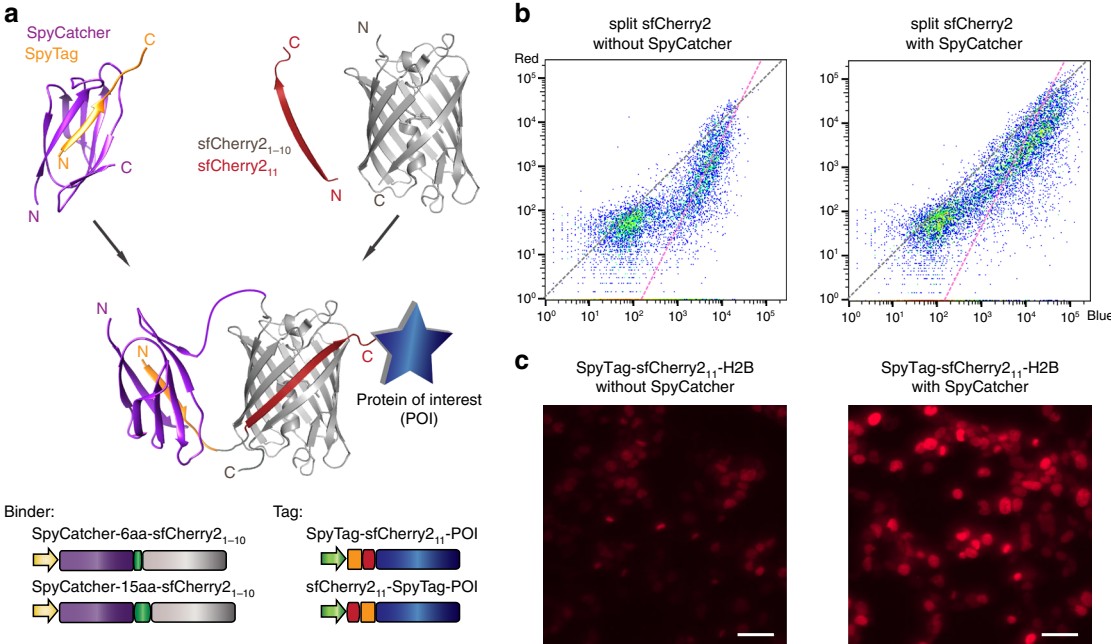

**Fig. 2** SpyTag/SpyCatcher-assisted complementation of split sfCherry2$_{1-10/11}$. **a** Schematic diagram of SpyTag/SpyCatcher-assisted complementation and construct design. **b** Flow cytometry analysis of whole-cell fluorescence in HEK 293T cells expressing SpyTag-sfCherry2$_{11}$-TagBFP with either sfCherry2$_{1-10}$ alone or with SpyCatcher-6aa-sfCherry2$_{1-10}$. The gray dashed trend line has a slope of 1 and the pink one has a slope of 2. **c** Fluorescence microscopy of sfCherry2$_{11}$ labeled H2B without or with the assistance of SpyTag/SpyCatcher interaction. The imaging condition and brightness/contrast range were set the same for better comparison. Scale bars: 50 μm

**Engineering split sfCherry3 for better complementation**. Previously, we engineered sfCherry2$_{1-10/11}$ using a spacer-insertion strategy[9]. This strategy was based on inserting a 32-aa spacer between the sfCherry$_{1-10}$ and sfCherry$_{11}$ coding regions. Beyond allowing mutagenesis of both fragments in a single PCR amplicon, the spatial constraints imposed by the linker were hypothesized to assist in the detection of the original mutations by raising the local concentrations of complementary fragments. To increase the screening stringency for complementation-enhancing mutations, we chose to express the fragments of sfCherry2 separately from two promoters using a pETDuet vector (Fig. 3a). Considering short peptides are prone to degradation in *Escherichia coli*, we fused the sfCherry2$_{11}$ sequence to the N-terminus of a well-folded carrier protein (SpyCatcher in our case).

We subjected the sfCherry2$_{1-10}$ fragment to four rounds of error-prone PCR mutagenesis and screening. In every round, a mixture of ~20 brightest variants were selected for the next round. The final isolated mutants were then subjected to one round of DNA shuffling. We have not mutated sfCherry2$_{11}$ so that all variants still bind the identical sfCherry2$_{11}$ peptide. In the end, sfCherry3C with five substitutions, K45R, G52D, T106A, K182R, N194D (numbering starts from the first Glu after the starting codon Met) was identified as the best variant after the directed evolution (Fig. 3b). All mutations were mapped to either surface orientated residues (T106A, K182R) or locations potentially interacting with the sfCherry2$_{11}$ peptide (K45R, G52D, and N194D).

To further improve the complementation efficiency of sfCherry3C, we introduced rational mutations inspired by a mCherry mutant named cp193g7 that is tolerant of circular permutations near our split site[18], because this mutant contains multiple similar mutations as in sfCherry[19] and sfCherry2 (ref. [9]). We combinatorially introduced the remaining mutations of cp193g7 (I7F, F65L, and L83W) into sfCherry3C through site-directed mutagenesis. Only the variant containing a single L83W

mutation gave brighter signal than sfCherry3C, which we designated as sfCherry3V (Fig. 3c). Complemented sfCherry3C$_{1-10/11}$ has an identical emission spectrum as that of sfCherry2 (both split and full-length), whereas the emission spectrum of complemented sfCherry3V$_{1-10/11}$ is blue-shifted by 5 nm (Fig. 3d). Fluorescence fluctuation spectroscopy indicates that sfCherry3C$_{1-10/11}$ has the same single-molecule brightness as sfCherry2 (Supplementary Fig. 5). On the other hand, sfCherry3V$_{1-10/11}$ is dimmer at the single-molecule level (Supplementary Fig. 5), which might be attributed to a difference in two-photon excitation of the blue-shifted chromophore at 1000 nm in the fluorescence fluctuation spectroscopy measurement.

Next, we performed the flow cytometry analysis (Fig. 3e) on HEK 293T cells co-transfected with mIFP-P2A-sfCherry2$_{11}$-Carrier and sfCherryX$_{1-10}$ (X being 2, 3C, or 3V). By fitting the data points (excluding those below a threshold above the scattering background) to a line with a fixed slope of 2 (Supplementary Fig. 6; see Supplementary Notes), we found a substantial up-shift of the fitted lines from sfCherry2 to sfCherry3C and sfCherry3V. The shifts corresponded to an increase of complemented signal by 2.5-fold and 8.2-fold, respectively. Assuming the same single-molecule brightness under one-photon excitation for flow cytometry, this signal enhancement also means the same fold of increase in the effective affinity between FP fragments.

We have further characterized other properties of the new split sfCherry3 variants that are important in practical applications. Split sfCherry3V demonstrated similar photobleaching rate as mCherry, whereas split sfCherry3C has a slightly slower photobleaching rate (Supplementary Fig. 7). pH stability is another factor can affect FP performance in specific situations. Split sfCherry3C showed a relative low pKa (5.0), close to that of mCherry (pKa ~ 4.8) which is known to be acid tolerant[20], whereas split sfCherry3V presented a higher pKa (5.9), making it still fully fluorescent at neutral pH but more sensitive to the acidic

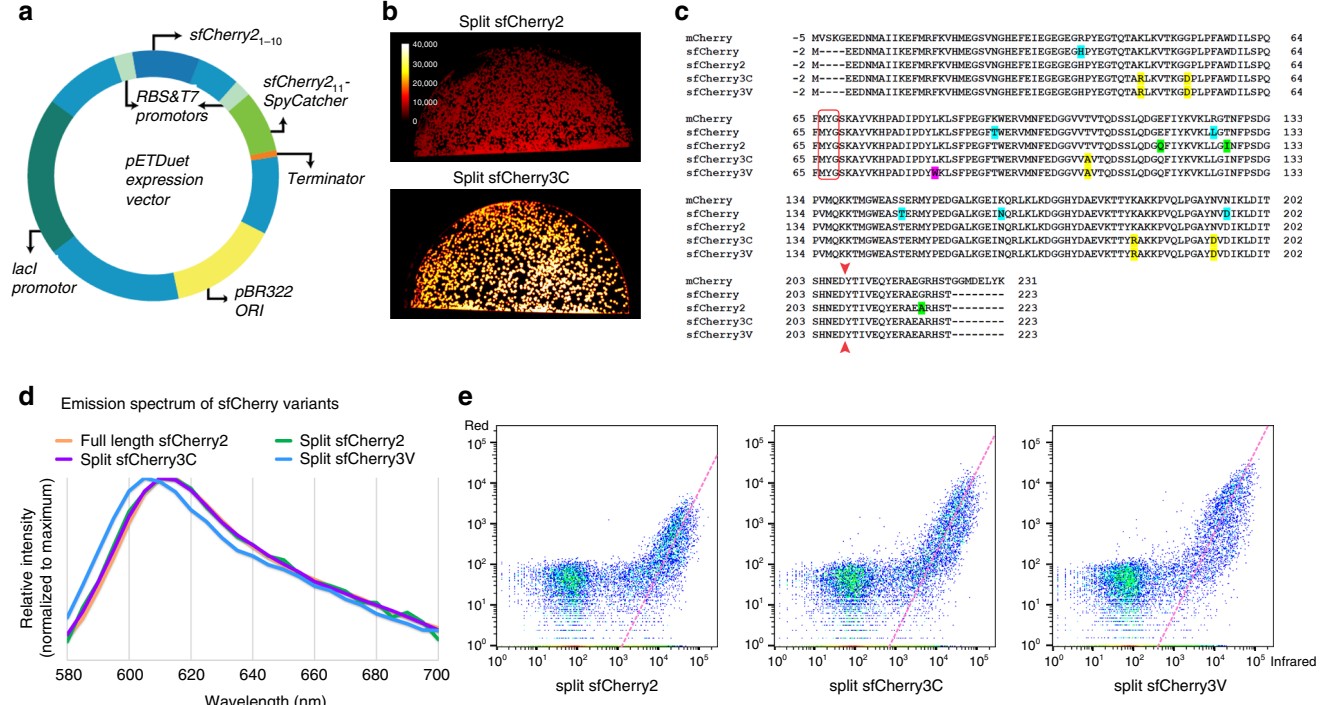

**Fig. 3** Engineering and characterization of split sfCherry3. **a** Schematics of pETDuet-based engineering platform. **b** Fluorescence images of *E. coli* colonies expressing split sfCherry2 or sfCherry3C from the pETDuet constructs. **c** Protein sequence alignment of mCherry, sfCherry, sfCherry2, sfCherry3C, and sfCherry3V. The amino acids forming the chromophore are indicated by a red box. The split site is indicated by the red arrow. Starting from mCherry, mutations introduced in sfCherry, sfCherry2, sfChery3C, and sfCherry3V are highlighted in cyan, green, yellow, and magenta, respectively. The overall alignment numbering follows that of sfCherry. **d** Emission spectra of sfCherry variants. **e** Flow cytometry analysis of whole-cell fluorescence in HEK 293T cells expressing mIFP-P2A-sfCherry$_{11}$-SpyCatcher and sfCherry2$_{1-10}$, sfCherry3C$_{1-10}$ or sfCherry3V$_{1-10}$. The pink dashed lines are results from linear least-square fitting with a fixed slope of 2 (see Supplementary Fig. 6)

cellular environment such as in the lumen lysosomes (Supplementary Fig. 8). This pH sensitivity is actually beneficial in tagging proteins in secretory pathways, which can suffer from fluorescence accumulation in lysosomes due to the inability for lysosomal protease to digest mCherry-derived fluorescent proteins[9]. Finally, we compared the temperature dependence of fluorescence in full length and split versions of sfCherry2, as well as split sfCherry3C, in *E. coli* cultures under either 37 or 25 °C (Supplementary Fig. 9). Although full-length sfCherry2 demonstrated similar fluorescence under different temperatures, the split sfCherry2 had lower intensity under the 37 °C. However, split sfCherry3C was able to recover the fluorescence in the physiological temperature 37 °C, which makes it advantageous for live-cell imaging applications.

**Endogenous protein labeling in human cells using sfCherry3.**
One unique application of FP$_{11}$ tag is to generate library-scale fluorescently labeled endogenous proteins through genetic knock-in by homology-directed DNA repair. The small 16-aa size of FP$_{11}$ allows us to fit its DNA sequence and short homology arms (~70 nt on either side) into commercially available 200 nt single-strand oligo-DNA (ssDNA). By electroporating Cas9/sgRNA ribonucleoprotein (RNP) and donor ssDNA into cells constitutively expressing the corresponding FP$_{1-10}$ fragment, robust generation of FP-labeled human cell lines becomes fast and cost-effective[4]. Multicolor knock-in has also been demonstrated by using orthogonal split FP systems to visualize differential distribution and interaction of multiple endoplasmic reticulum proteins[9].

The overall increased brightness of complemented sfCherry3 variants make them superior to sfCherry2 in the application of labeling endogenous proteins through knock-in. Because sfCherry3 and sfCherry2 share the same FP$_{11}$ fragment, we adopted a reversed strategy as our previously reported one: knock-in of sfCherry2$_{11}$ into HEK 293T wild-type cells through electroporation, followed by lentivirus infection for the three sfCherry$_{1-10}$ variants (schematics in Fig. 4a). A total of six sfCherry2$_{11}$ cell lines were created, with knock-ins at: lamin A/C (LMNA, inner nuclear membrane), clathrin light chain A (CLTA), RAB11A, heterochromatin protein 1 β (HP1b), endoplasmic reticulum proteins SEC61b (translocon complex), and ARL6IP1 (tubular ER). We compared fluorescence-activated cell sorting (FACS) enrichment efficiency for each cell line after infection with lentivirus (Fig. 4b–g). In all examined targets, the sfCherry3C$_{1-10}$ and sfCherry3V$_{1-10}$ groups displayed remarkable population enhancement in the red-fluorescence-positive gate compared to sfCherry2$_{1-10}$, rendering the sorting process substantially faster. Practically, for targets like CLTA, SEC61b, or ARL6IP1, we were able to gate the fluorescent population around the clear peak and have 5–10-fold higher yield of isolating cells with successful knock-in.

We further confirmed our knock-ins were on-target through confocal microscopy imaging (Fig. 4b–g). Consistent with its higher p*K*a, sfCherry3V$_{1-10}$ groups have a reduced tendency to show fluorescent puncta from lysosomes (Fig. 4g) which is reported in our previous work[9]. This makes the sfCherry3V$_{1-10}$ the preferred protein fragment when labeling endogenous proteins involved in the endomembrane system.

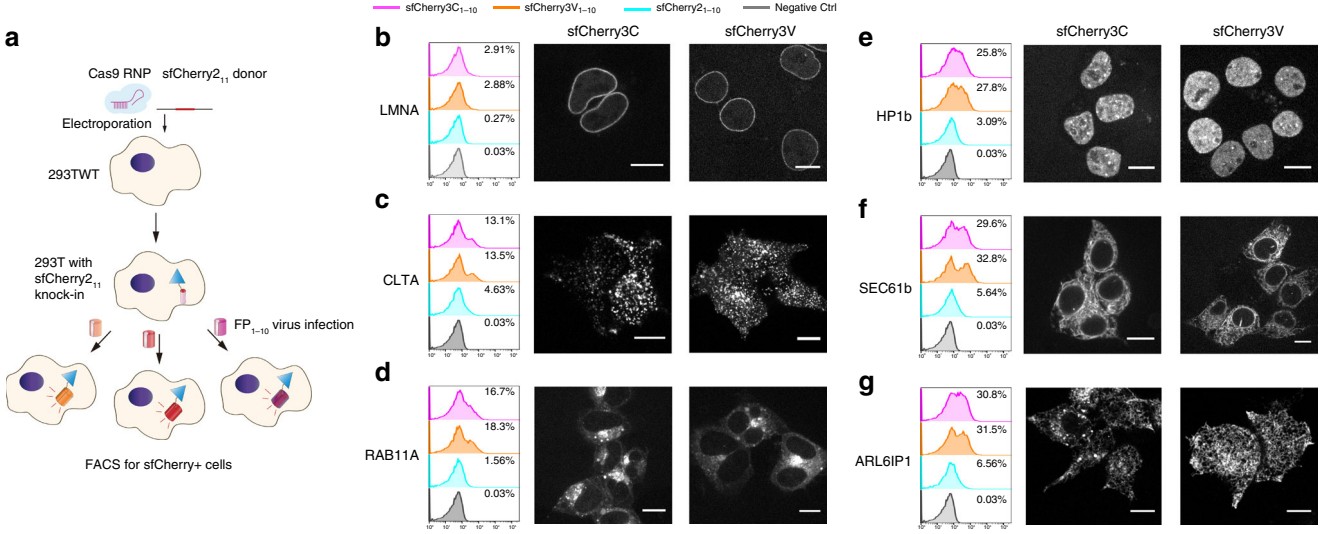

**Fig. 4** Endogenous protein labeling in HEK 293T cells using sfCherry3 variants. **a** Schematic diagram of knock-in followed by virus infection and FACS enrichment. **b–g** Analysis of FACS sorting efficiency in six targets, **b** lamin A/C, **c** clathrin light chain A, **d** RAB11A, **e** heterochromatin protein 1 β, **f** ER translocon complex SEC61b, and **g** ER tubule protein ARL6IP1, and visualization of sorted knock-in cells through confocal fluorescence microscopy. Scale bar: 10 μm

**NLG-1 CLASP visualizes specific synapses in live animals.** To visualize synapses between specific sets of pre- and postsynaptic neurons in live animals, the trans-synaptic marker NLG-1 GRASP was designed using split GFP fragments[8]. Complementary split GFP$_{1–10/11}$ fragments were linked via a flexible linker to the transmembrane synaptic protein Neuroligin, which localizes to both pre- and postsynaptic sites in *Caenorhabditis elegans*[8]. When the two neurons in which the complementary pre- and postsynaptic markers are expressed form synapses, the split GFP fragments come into contact, reconstitute, and fluoresce (Fig. 5a). Using NLG-1 GRASP, we discovered that the recognition between two synaptic partners, the PHB sensory neurons and the AVA interneurons, is mediated the secreted ligand UNC-6/Netrin, its canonical receptor UNC-40/Deleted in Colorectal Cancer[21], and the receptor protein tyrosine phosphatase (RPTP) CLR-1[13]. NLG-1 GRASP has also been adapted to many other systems, indicating that this technology is transferable[22]. The addition of a red fluorescent trans-synaptic marker would greatly expand this system, allowing us to differentially label different subsets of synapses within the same animals, and potentially even within the same neuron.

Leveraging the split-sfCherry3 tagging system, we have developed the split sfCherry3-based NLG-1 CLASP. The left and right PHB sensory neurons, located in the posterior of *C. elegans*, form the majority of their synapses with the left and right AVA and PVC interneurons[11,12]. In this study, we focused on synapses formed between the two PHB neurons with the two AVA interneurons. To visualize PHB-AVA synapses with this newly developed split sfCherry3C, constructs were generated in which the sequence encoding the large fragment sfCherry3C$_{1–10}$ was linked to the N-terminus of the *neuroligin-1 (nlg-1)* cDNA after the *nlg-1* signal sequence via a flexible 12GS linker. This half of the marker was expressed in PHB neurons using a promoter that within the posterior half of the worm is specific for these presynaptic neurons ($_p$gpa-6)[23]. The complementary small fragment sfCherry3C$_{11}$ was similarly linked to *nlg-1* and expressed in AVA neurons using a promoter that within the posterior half of the worm is specific for these postsynaptic neurons ($_p$flp-18)[24]. Transgenic lines carrying both $_p$gpa-6::nlg-1::sfCherry3C$_{1–10}$ and $_p$flp-18::nlg-1::sfCherry3C$_{11}$ were generated. In wild-type animals, the distribution of red fluorescent puncta was

similar to those observed in animals labeled with PHB-AVA NLG-1 GRASP (Fig. 5), and to that described by serial electron microscopy reconstruction[11,12]. To determine if these puncta were indeed synaptic, we introduced NLG-1 CLASP into *clr-1/RPTP* synaptogenesis mutants. In *clr-1/RPTP* mutants, PHB-AVA NLG-1 GRASP fluorescence intensity is dramatically reduced, and a PHB circuit-specific behavior is disrupted, indicating a reduction in synaptogenesis between the two neurons[13]. However, contact between the PHB and AVA neurons remains intact[13]. We found that NLG-1 CLASP fluorescence intensity was also dramatically reduced in *clr-1/RPTP* mutants (Fig. 5), further indicating that NLG-1 CLASP puncta are synaptic, and that NLG-1 CLASP has very little, if any background fluorescence.

We further set out to determine if NLG-1 CLASP could be used in combination with NLG-1 GRASP to differentially label synapses within the same neuron that were made with different neuronal partners. We focused on AVA command interneurons, with the goal of labeling synapses between PHB and AVA neurons with NLG-1 CLASP, and synapses between AVA neurons and their postsynaptic partner, the VA and DA motorneurons, which are spaced throughout the ventral nerve cord. We generated PHB-AVA NLG-1 CLASP constructs with sfCherry3V and generated transgenic animals carrying these as well as AVA-VA/DA NLG-1 GRASP[8,13]. Previous work demonstrates that increasing the number of synapses between PHB and AVA neurons results in potentiating the response to the PHB circuit-sensed chemical sodium dodecyl sulfate (SDS)[13,21]. However, PHB circuit function in animals carrying PHB-AVA NLG-1 CLASP as well as AVA-VA/DA NLG-1 GRASP was normal, indicating that NLG-1 CLASP did not induce additional synaptogenesis between PHB and AVA neurons (Supplementary Fig. 10). Animals carrying PHB-AVA NLG-1 CLASP and AVA-VA/DA NLG-1 GRASP had synaptic patterns consistent with previous electron microscopy and fluorescent labeling (Fig. 6)[8,11–13,21]. While NLG-1 CLASP-labeled synapses between PHB and AVA were present where PHB axons extend (in the preanal ganglion), synapses between AVA and VA and DA interneurons stretched from this region anteriorly along the ventral nerve cord where VA and DA neurites are found. *clr-1/RPTP* is required for both of these sets of synapses[13]. Therefore, to test if these two markers correctly labeled synapses, we introduced

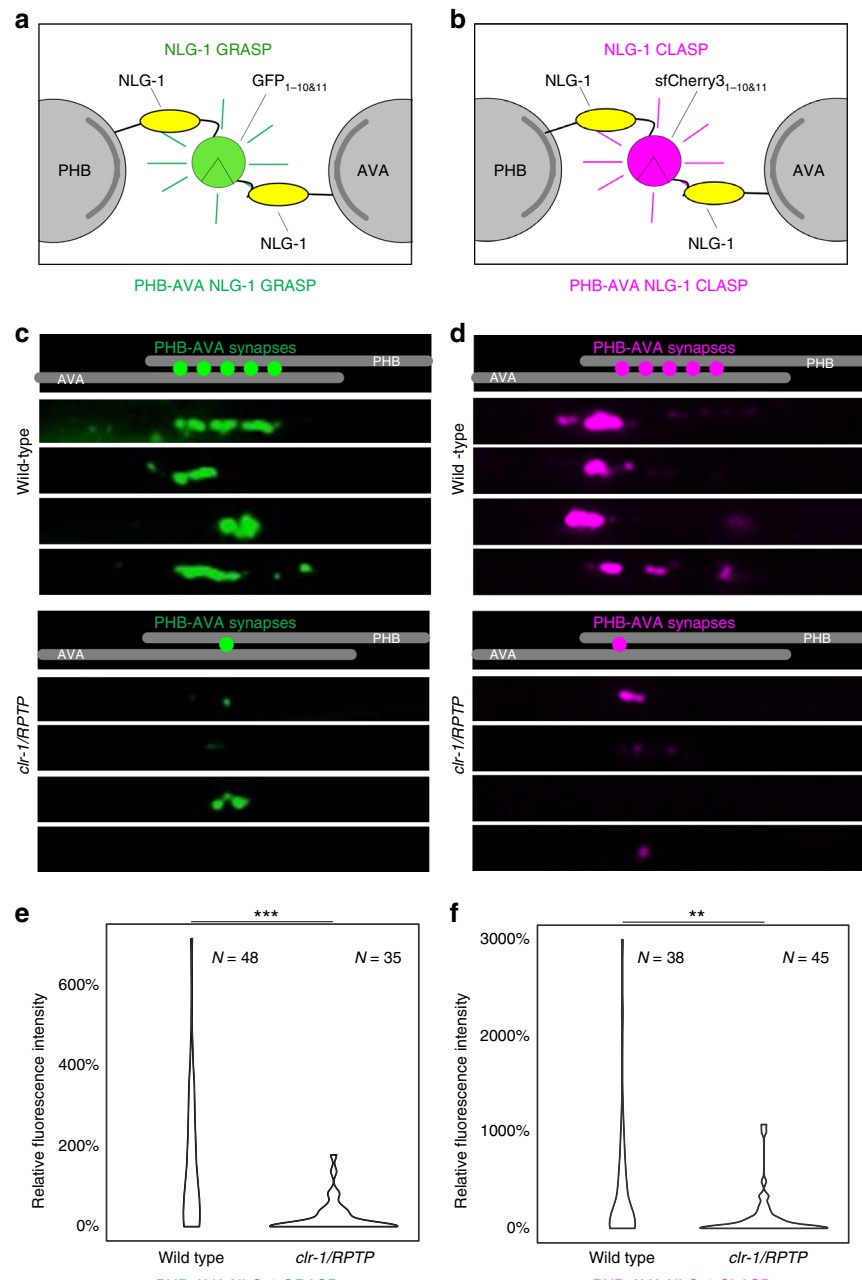

**Fig. 5** NLG-1 CLASP visualizes specific subsets of synapses in live *C. elegans*. **a**, **b** Schematic diagrams of GFP$_{1-10/11}$-based NLG-1 GRASP and sfCherry3$_{1-10/11}$-based NLG-1 CLASP expressed in PHB and AVA neurites. **c**, **d** Schematics and micrographs of NLG-1 GRASP and NLG-1 CLASP specifically labeling synaptic contacts between the two PHB and the two AVA neurons in *C. elegans* (one neuron from each pair is shown for simplicity). PHB-AVA NLG-1 GRASP and NLG-1 CLASP fluorescence intensity are dramatically reduced in the synaptic partner recognition mutant *clr-1/RPTP*. **e**, **f** Quantification of the reduction in relative fluorescence intensities of NLG-1 GRASP and NLG-1 CLASP in *clr-1/RPTP* mutants. $N =$ animals in each case, ***$P < 0.001$, **$P < 0.01$, Mann–Whitney $U$-test. See Supplementary Data for the list of data values

the *clr-1/RPTP* loss-of-function mutation into the PHB-AVA NLG-1 CLASP and AVA-VA/DA NLG-1 GRASP transgene-carrying animals. We found that fluorescence of both markers was severely reduced, consistent with these markers being synaptic.

## Discussion
In this study, we have developed bright sfCherry3C$_{1-10/11}$ (and its variant sfCherry3V$_{1-10/11}$) with enhanced complementation efficiency, enabling the high-efficient generation of human cell lines with endogenously sfCherry-labeled proteins. Moreover, we have transformed the split sfCherry3 into a trans-synaptic marker called NLG-1 CLASP and have demonstrated the visualization of specific synapses in living nervous systems.

The proposed two-step model, consisting of a dynamic association/dissociation equilibrium followed by an irreversible folding/maturation process, can be generalized to other split fluorescent proteins, including the non-self-associating ones used to monitor protein–protein interaction in bimolecular fluorescence complementation (BiFC) assays[25,26]. In fact, there is no definitive boundary between the non-self-associating and self-associating split FPs. Instead, their only difference is in the

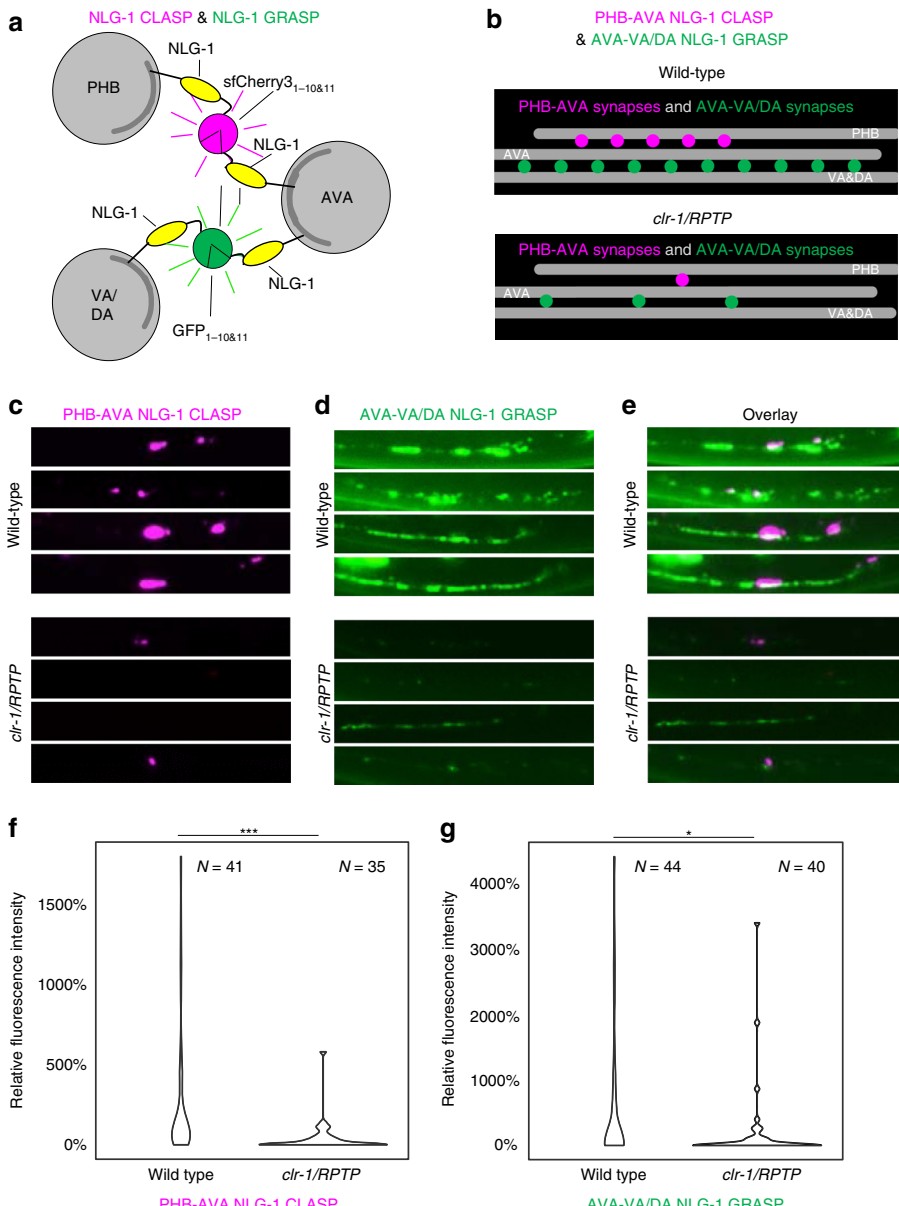

**Fig. 6** NLG-1 CLASP and NLG-1 GRASP distinguish synapses made between the AVA neurons and two different sets of synaptic partners in live *C. elegans*. **a** Schematic diagram of sfCherry3$_{1-10/11}$-based NLG-1 CLASP labeling synapses between the two AVA interneurons and their presynaptic partners, the PHB sensory neurons, and GFP$_{1-10/11}$-based NLG-1 GRASP labeling synapses between the two AVA interneurons and their postsynaptic partners, the VA and DA motorneurons (one neuron from each group is shown for simplicity). **b–e** Schematics and micrographs of NLG-1 CLASP specifically labeling synaptic contacts between AVA and PHB neurons, and NLG-1 GRASP specifically labeling synaptic contacts between AVA and VA and DA neurons. Each row contains micrographs of the same animal. PHB-AVA NLG-1 CLASP and AVA-VA/DA NLG-1 GRASP fluorescence intensity are dramatically reduced in the synaptic partner recognition mutant *clr-1/RPTP*. **f, g** Quantification of the reduction in relative fluorescence intensities of NLG-1 CLASP and NLG-1 GRASP in *clr-1/RPTP* mutants. N = animals in each case, \*\*\*P < 0.001, \*P < 0.05, Mann–Whitney *U*-test. See Supplementary Data for the list of data values

spontaneous binding affinity, which can be characterized by $K'_D$ in our model. For BiFC analysis, this affinity is one of the major determinant of the sensitivity. If it is too low, the split FP will fail to produce sufficient complementation signal even when the probed molecular interaction does occur. On the other hand, the opposite extreme leads to high background complementation as observed in certain split constructs[6]. While this $K'_D$ is not straightforward to measure biochemically in vitro due to the overall irreversible nature of complementation, our steady-state flow cytometry analysis provides a reliable way to characterize it in cells. Our model also shows that the overall complemented signal is determined not only thermodynamically by the initial binding affinity and local fragment concentrations, but also

kinetically by the rates of folding, chromophore maturation and protein degradation.

The assisted complementation demonstrated by SpyTag/Spy-Catcher presents a simple way to improve complementation efficiency between FP fragments. This strategy can be expanded to multicolor imaging using FP$_{11}$ tagging using orthogonal binder/tag pairs such as SnoopTag/SnoopCatcher (engineered by splitting an adhesin from *Streptococcus pneumoniae*[27]) or SsrA/SspB (a degradation tag and its adaptor protein from bacterial ClpX ATPase[28,29]).

The substantial improvement of complementation efficiency in sfCherry3$_{1-10/11}$ is beneficial for a wide variety of applications ranging from protein labeling to scaffolding protein complexes, as

well as monitoring cell–cell connections. A highly efficient complementation process not only guarantees an enhanced overall brightness, but also enables us to tune down the expression level of $FP_{1-10}$ fragments which might otherwise exhaust the cellular machinery that maintaining the protein homeostasis (biogenesis, folding, trafficking, and degradation of proteins). This benefit is essential in scenarios that are sensitive to the expression of exogenous proteins, such as tagging endogenous proteins in embryos. Moreover, our engineering platform based on pETDuet vector can be utilized to optimize other self-associating split FPs with insufficient complementation efficiency, such as $mNG2_{1-10/11}$.

Utilizing the split $sfCherry3_{1-10/11}$ construct, we have established a completely orthogonal, red-colored trans-synaptic marker called NLG-1 CLASP. We demonstrate that this marker can be used in combination with the original green fluorescent trans-synaptic marker NLG-1 GRASP to label subsets of synapses made between a neuron and different synaptic partners. Similar cyan and yellow trans-synaptic markers (called dual-eGRASP) have recently been generated in vertebrates[30]. The use of NLG-1 CLASP with these cyan and yellow trans-synaptic markers has the potential to allow simultaneous and differential labeling of synapses between a single neuron and three synaptic partners in live animals. Since many synaptic connections within the nematode have overlapping localizations within the nerve ring and other nerve bundles[11,12], this tool will allow accurate visualization of multiple subsets of a single neuron's connections. These tools may similarly be of use in densely innervated regions of the vertebrate nervous system, such as the hippocampus and cortex. Thus, we propose that NLG-1 CLASP will be a powerful tool with which to probe the development and plasticity of neural circuits within live animals.

Taken together, our work greatly expands the SAsFP toolbox, not only by providing a new, red SAsFP with improved complementation signal, but also by mechanistic elucidation of the complementation process to laid down rational engineering routes. These advancements have led to a drastic efficiency improvement in red-tagging of endogenous proteins via gene editing to systematically visualize protein–protein colocalization and interactions. Moreover, they also create a completely orthogonal color channel for the detection of neuronal synapses, allowing simultaneous and differential labeling of synapses between multiple synaptic partners in live animals, paving the way for mapping complex neuronal connectivity networks, and understanding the genes and environmental factors that shape and influence neuronal connectivity.

## Methods

**Molecular cloning**. The DNA sequence of SpyCatcher 002 and SpyTag 002 (based on the reported sequence from ref. [17]) were directly synthesized (Integrated DNA Technologies, IDT). The DNAs of histone H2B, TOMM20, TagBFP, and mIFP were subcloned from mEmerald, TagBFP, or mIFP fusion plasmids (cDNA source: the Michael Davidson Fluorescent Protein Collection at the UCSF Nikon Imaging Center) using Phusion High-Fidelity DNA Polymerase (Thermo Scientific). The P2A sequence are GCTACTAACTTCAGCCTGCTGAAGCAGGCTGGA-GACGTGGAGGAGAACCCTGGACCT. The lentiviral plasmids pSFFV-GFP$_{1-10}$, pSFFV-mNG2$_{1-10}$, and pSFFV-sfCherry2$_{1-10}$ were generated in our previous research[9]. To build three DNA fragments encoding mIFP, P2A-FP$_{11}$, and SpyCatcher were ligated into linearized pSFFV vector (BamHI/NotI) using In-Fusion HD Cloning kit (Clontech) within one reaction. To construct pSFFV-TagBFP-P2A-sfCherry2$_{1-10}$ plasmid (used in Fig. 1d), two DNA fragments encoding TagBFP or P2A were ligated in to linearized pSFFV-sfCherry2$_{1-10}$ vector using In-fusion. To generate pcDNA-SpyCatcher-6aa-sfCherry2$_{1-10}$ and pcDNA-SpyCatcher-15aa-sfCherry2$_{1-10}$ plasmids (used in Fig. 2a, b), PCR amplicons encoding SpyCatcher or sfCherry2$_{1-10}$ were cloned into digested pcDNA3.1 vectors (HindIII/BamHI), and the different linkers were achieved through designing overlapping primers with various linker lengths. To make pSFFV-SpyTag-sfCherry2$_{11}$-TagBFP, pSFFV-SpyTag-sfCherry2$_{11}$-TagBFP-H2B (used in Fig. 2b, c), pmEmerald-TOMM20-sfCherry2$_{11}$-SpyTag, DNA fragments encoding SpyTag-sfCherry2$_{11}$, TagBFP and

H2B, or sfCherry2$_{11}$-SpyTag were ligated in to either linearized pSFFV vector (BamHI/NotI) or linearized pmEmerald (AgeI/NotI) through In-Fusion.

The pETDuet-1 vector was kindly donated by Dr. Alexander Kintzer from Dr. Robert Stroud's laboratory at UCSF. We generated the initial plasmid for mutagenesis screening by two rounds of In-Fusion ligation reaction: inserting the PCR amplicon encoding sfCherry2$_{1-10}$ into the first multiple cloning site (MCS) of pETDuet-1 digested by NcoI, followed by inserting the DNA sequence encoding sfCherry2$_{11}$-SpyCatcher into the second MCS of pETDuet-1 digested by NdeI. The NcoI restriction site preserved in the final product, but the NdeI restriction site was destroyed. For the mammalian expression and lentiviral production, DNAs of sfCherry3C$_{1-10}$ were directly PCR amplified from identified pETDuet-1 construct (final mutant) and cloned into the lentiviral pSFFV vector (BamHI/NotI). To generate the sfCherry3V$_{1-10}$ variant, we introduced the point mutation L83W into pSFFV-sfCherry3C$_{1-10}$ plasmid using QuikChange II Site-Directed Mutagenesis Kit (Agilent Technologies). For the complete nucleotide sequence of sfCherry3C$_{1-10}$, sfCherry3V$_{1-10}$, SpyCatcher 002, and SpyTag 002, see Supplementary Table 1.

Constructs used in the NLG-1 CLASP application were generated using standard molecular techniques. To generate $_pgpa-6::nlg-1::sfCherry3C_{1-10}$ construct (MVC227), sfCherry3C$_{1-10}$ was amplified from $pSFFV-sfCherry3C_{1-10}$ using the following primers: MVP846 (AGCTGCTAGCATGGAACGCATTTATCTTCTT CTCCTTCTTTTTCTGCCCAGGATACGATCCATGGAGGAGGACAACATGG) and MVP847 (TCCGGAGCTCGTCCTCGTTGTGGCTGGT). The fragment was subcloned into $_pgpa-6::nlg-1::GFP_{1-10}$ (MVC6)[21], replacing GFP$_{1-10}$, using the NheI and SacI sites. To generate the $_pflp-18::nlg-1::sfCherry2_{11}$ (MVC228) construct, sfCherry2$_{11}$ was amplified from $H_2B-sfCherry2_{11}$ using the following primers: MVP848 (AGCTGCTAGCATGGAACGCATTTATCTTCTTCTCCTTCTTTTT TCTGCCCAGGATACGATCCTACACCATCGTGGAGCAGT) and MVP849 (TCCGGAGCTCGGTGCTGTGTCTGGCCTC). The fragment was subcloned into $_pflp-18::nlg-1::GFP_{11}$ (MVC12)[21], replacing GFP$_{11}$, using the NheI and SacI sites. To generate $_pgpa-6::nlg-1::SFCherry_{3V1-10}$ (MVC233) construct, site-directed mutagenesis (SDM) was performed using the QuikChange II Site-Directed Mutagenesis Kit on $_pgpa-6::nlg-1::sfCherry3C_{1-10}$ (MVC227) using following primers: MVP861 (GGGGAAGCTCAGCTTCCAGTAGTCGGGGATGTCG) and MVP862 (CGACATCCCCGACTACTGGAAGCTGAGCTTCCCC).

**Cell culture and lentiviral production**. Human HEK 293T cells (UCSF cell culture facility. Not authenticated. Tested and identified no mycoplasma contamination.) were maintained in Dulbecco's modified Eagle's medium (DMEM) with high glucose (Gibco), supplemented with 10% (vol/vol) FBS and 100 μg ml$^{-1}$ penicillin/streptomycin (UCSF Cell Culture Facility). U2OS cells (American Type Culture Collection, Manassas, VA) were cultured in DMEM media, supplemented with 10% fetal bovine serum. All cells were grown at 37 °C and 5% CO$_2$ in a humidified incubator. For the lentiviral production, $1 \times 10^6$ HEK 293T cells were plated into T25 one day prior to transfection. Four hundred and thirty nanograms of pMD2.G plasmid, 3600 ng of pCMV-dR8.91 plasmid, and 4100 ng of the lentiviral plasmid (pSFFV-sfCherry2$_{1-10}$, pSFFV-sfCherry3C$_{1-10}$ and pSFFV-sfCherry3V$_{1-10}$) were co-transfected into HEK 293T cells using FuGENE HD (Promega) following the manufacturer's recommended protocol. The virus containing supernatant is harvested 48 h after transfection and were centrifuged to pellet any packaging cells. Virus containing medium is used immediately or stored in −80 °C freezer for future use.

For single-molecule brightness measurement sample preparation, U2OS cells were grown in 24-well plates with #1.5 glass coverslip bottoms (In Vitro Scientific) and transfected ~24 h before measurement using GenJet transfection reagent (SignaGen Laboratories) according to the manufacturer's instructions. Immediately before measurement, the growth media was exchanged with PBS buffer with calcium and magnesium (Gibco).

**Sample preparation and data analysis in flow cytometry**. To characterize the relationship between complementation efficiency and expression level in split GFP, split mNG2, and split sfCherry2, we made pSFFV-mIFP_P2A_full-length FP and pSFFV-mIFP_P2A_FP$_{11}$-SpyCatcher constructs. Corresponding to each scatter plots in Fig. 1b–d, $3 \times 10^4$ HEK 293T cells grown on a 48-well plate (Eppendorf) were co-transfected with (B) left: 100 ng pSFFV-mIFP_P2A_GFP$_{11}$-SpyCatcher with 200 ng pSFFV-GFP$_{1-10}$, right: 100 ng pSFFV-mIFP_P2A_GFP[full-length] with 200 ng pSFFV-GFP$_{1-10}$; (C) left: 100 ng pSFFV-mIFP_P2A_mNG2$_{11}$-Spy-Catcher with 200 ng pSFFV-mNG2$_{1-10}$, right: 100 ng pSFFV-mIFP_P2A_mNG2 [full-length] with 200 ng pSFFV-mNG2$_{1-10}$; (D) left: 100 ng pSFFV-mIFP_P2A_sfCherry2$_{11}$-SpyCatcher with 200 ng pSFFV-sfCherry2$_{1-10}$, right: 100 ng pSFFV-mIFP_P2A_sfCherry2[full-length] with 200 ng pSFFV-sfCherry2$_{1-10}$. In Fig. 1e, same cells were co-transfected with 100 ng pSFFV-mIFP_P2A_sfCherry2$_{11}$-SpyCatcher and 200 ng pSFFV-sfCherry2$_{1-10}$-TagBFP.

To test the Spy002 pair assisted complementation in sfCherry2, we built two tandem-binder constructs (pcDNA-SpyCatcher-6aa-sfCherry2$_{1-10}$ and pcDNA-SpyCatcher-15aa-sfCherry2$_{1-10}$) and two tandem-tag constructs (pSFFV-SpyTag-sfCherry2$_{11}$-TagBFP and pSFFV-sfCherry2$_{11}$-SpyTag-TagBFP). We then combinatorially co-transfected the same cells with 100 ng tag construct plus 200 ng binder construct. In Fig. 2b, to compare the complementation efficiency with or without Spy pair interaction, cells were co-transfected with either 100 ng pSFFV-

SpyTag-sfCherry2$_{11}$-TagBFP plus 200 ng pcDNA-sfCherry2$_{1-10}$, or 100 ng pSFFV-SpyTag-sfCherry2$_{11}$-TagBFP plus 200 ng SpyCatcher-6aa-sfCherry2$_{1-10}$.

To validate the increased complementation of split sfCherry3 variants versus split sfCherry2, we made pSFFV-sfCherry3C$_{1-10}$ and pSFFV-sfCherry3V$_{1-10}$ constructs. Corresponding to each scatter plots in Fig. 3e, same cells were co-transfected using with (left) 100 ng pSFFV-mIFP_P2A_sfCherry2$_{11}$-SpyCatcher with 200 ng pSFFV-sfCherry2$_{1-10}$; (middle) 100 ng pSFFV-mIFP_P2A_sfCherry2$_{11}$-SpyCatcher with 200 ng pSFFV-sfCherry3C$_{1-10}$; (right) 100 ng pSFFV-mIFP_P2A_sfCherry2$_{11}$-SpyCatcher with 200 ng pSFFV-sfCherry3V$_{1-10}$.

For flow cytometry analysis, 48 h after transfection, transfected HEK 293T cells were digested with Trypsin-EDTA (0.25%) (Gibco) into single cells and resuspended in 0.5 ml PBS solution. Analytical flow cytometry was carried out on a LSR II instrument (BD Biosciences) and cell sorting on a FACSAria II (BD Biosciences) in Laboratory for Cell Analysis at UCSF. Flow cytometry data analysis (gating by the scattering channel and plotting) was conducted using the FlowJo software (FlowJo LLC).

**Single-molecule brightness measurement and data analysis.** Fluorescence brightness measurements were carried out on a homebuilt two-photon microscope, which has been previously described[31]. Pulsed laser light (100 fs pulses with a repetition frequency of 80 MHz) from a mode-locked Ti-Sapphire laser (Mai-Tai, Spectra Physics) was focused by a ×63 C-Apochromat water immersion objective (NA = 1.2, Zeiss) to create two-photon excitation. The emitted fluorescence was collected by the same objective and separated from the excitation light by a dichroic mirror (675DCSXR, Chroma Technology). The fluorescence emission was separated into two detection channels by a 580 nm dichroic mirror (585DCXR, Chroma Technology), and the green channel was further filtered by an 84nm-wide bandpass filter centered at 510 nm (FF01-510/84-25, Semrock). Avalanche photodiodes (SPCM-AQ-14, Perkin-Elmer) detected the fluorescence signal, and photon counts were recorded by a data acquisition card (FLEX02, Correlator.com) for ~60 s with a sampling frequency of 200 kHz. All measurements were carried out at an excitation wavelength of 1000 nm and a measured power after the objective of ~0.46 mW. The photon count record was analyzed to recover Mandel's Q parameter as previously described[32] using programs written for IDL 8.7 (Research Systems, Inc.). The Q value is converted into brightness $\lambda$, which represents the average fluorescence intensity per molecule, using the relation $Q = \gamma_2 \lambda T$, where $T$ represents the sampling time and $\gamma_2$ is a shape-dependent factor whose value has been determined as described previously[33].

**Mutagenesis and screening.** The amino-acid sequence of sfCherry2 and the split site were from our previous published literature[9]. The sfCherry2$_{1-10}$ sequence was subjected to random mutagenic PCR (forward primer: AGGAGATATACCATG GAGGAGGACAAC, reverse primer: CTGCTGCCCATGTCAGTCCTCGTTGTG) using the GeneMorph II Random Mutagenesis Kit (Agilent Technologies). A high mutation rate protocol suggested in the instruction manual was adapted, with an initial target DNA amount of 0.2 μg and 30-cycle amplification. The cDNA library pool was gel-purified and ligated into a PCR-linearized pETDuet vector (forward primer: TGACATGGGCAGCAGCCA, reverse primer: CATGGTATATCTCCTT CTTAAAGTTAAACAAAATTATTTCTAGAGG. The product only contains the sfCherry2$_{11}$-SpyCatcher coding sequence in the second MCS but not the sfCherry2$_{1-10}$ in the 1firstMCS.) using In-Fusion. The plasmid pool was then transformed into E. coli BL21 (DE3) electrocompetent cells (Lucigen) by electro-poration using the Gene Pulser Xcell™ Electroporation Systems (Bio-Rad). The expression library was evenly plated on nitrocellulose membrane (Whatman, 0.45 μm pore size), which was sitting on an LB-agar plate with 30 mg ml⁻¹ kanamycin. After overnight growth at 37 °C, the nitrocellulose membrane was carefully transferred onto a new LB-agar plate containing 1 mM isopropyl-β-D-thioga-lactoside (IPTG) and 30 mg ml⁻¹ kanamycin and cultured for another 3–6 h at 37 °C to induce the protein production. We performed the clone screening by imaging the IPTG-containing LB-agar plate using a BioSpectrum Imaging System (UVP). The brightest candidates in each library were pooled (typically ~20 from approximately 10,000 colonies) and served as templates for the next round of directed evolution. The DNA sequences of selected constructs were confirmed by sequencing (Quintara Biosciences). For DNA shuffling, we adopted the protocol described in Yu et al.[34]. Specifically, we PCR amplified the brightest six sfCherry3$_{1-10}$ variants (forward primer: AGGAGATATACCATGGAGGAGGAC AAC, reverse primer: CTGCTGCCCATGTCAGTCCTCGTTGTG) from the last round of random mutagenesis. PCR products of 651 bp were purified from 1% agarose gels using zymoclean gel DNA gel recovery kit (Zymo Research). The DNA concentrations were measured in Nanodrop and the fragments were mixed at equal amounts for a total of ~2 μg. The mixture was then digested with 0.5 unit DNase I (New England Biolabs) for 13 min and terminated by heating at 95 °C for 10 min. The DNase I digests were run on a 2% agarose gel, and the band with a size of 50–100 bp was selected and purified. Ten microliters of purified fragments was added to 10 μl of Phusion High-Fidelity PCR Master Mix and reassembled with a PCR program of 30 cycles, with each cycle consisting of 95 °C for 60 s, 50 °C for 60 s, and 72 °C for 30 s. After reassembly, 1 μl of this reaction was amplified by PCR. The shuffled library was then transformed, expressed, and screened as described above. After the directed evolution was saturated (no apparent

fluorescence increase in the induced colonies), the brightest clone was selected and the DNA sequences of the constructs were confirmed by sequencing (Quintara Biosciences). The emission spectra of split sfCherry variants (in E. coli solution culture expressing FPs) were measurement on a Synergy™ H4 Hybrid Multi-Mode Microplate Reader from UCSF Center for Advanced Technology.

**CRISPR/Cas9-mediated gene editing.** We purchased synthetic single-strand DNA oligos from Integrated DNA Technologies (IDT). We prepared sgRNAs and Cas9/sgRNA RNP complexes following our published methods[4]. Specifically, sgRNAs were obtained by in vitro-transcribing DNA templates containing a T7 promoter (TAATACGACTCACTATAG), an sgRNA scaffold region, and the gene-specific 20 nt sgRNA sequence. DNA templates were produced by overlapping PCR using a set of four primers: three common primers (forward primer T25: 5′-TAA TAC GAC TCA CTA TAG-3′; reverse primer BS7: 5′-AAA AAA AGC ACC GAC TCG GTG C-3′; and reverse primer ML611: 5′-AAA AAA AGC ACC GAC TCG GTG CCA CTT TTT CAA GTT GAT AAC GGA CTA GCC TTA TTT AAA CTT GCT ATG CTG TTT CCA GCA TAG CTC TTA AAC-3′) and one gene-specific primer (forward primer 5′-TAA TAC GAC TCA CTA TAG NNN NNN NNN NNN NNN NNN NNG TTT AAG AGC TAT GCT GGA A-3′). For each template, a 50 μL PCR was performed with Phusion® High-Fidelity PCR Master Mix (New England Biolabs) reagents with the addition of 1 μM T25, 1 μM BS7, 20 nM ML611, and 20 nM gene-specific primer. The PCR product was purified and eluted in 12 μL of RNAse-free DNA buffer. Next, a 100-μL in vitro transcription reaction was performed with ~300 ng DNA template from PCR product and 1000 U of T7 RNA polymerase in buffer containing 40 mM Tris pH 7.9, 20 mM MgCl₂, 5 mM DTT, 2 mM spermidine and 2 mM of each NTP (New England BioLabs). Following a 4 h incubation at 37 °C, the sgRNA product was purified and eluted in 15 μL of RNAse-free RNA buffer. The sgRNA was quality-checked by running 5 pg of the product on Mini-PROTEAN TBE Precast Gels (Bio-Rad Laboratories) at 200 V for 60–80 min.

For the knock-in of sfCherry2$_{11}$ into HEK 293T wild-type (WT) cells, 200 nt homology-directed recombination (HDR) templates were ordered in single-stranded DNA (ssDNA) from IDT. For the complete set of DNA sequence used for sgRNA in vitro transcription or HDR templates, see Supplementary Tables 2 and 3. Cas9 protein (pMJ915 construct, containing two nuclear localization sequences) was expressed in E. coli and purified by the University of California Berkeley Macrolab. HEK 293T WT cells were treated with 200 ng ml⁻¹ nocodazole (Sigma) for ~17 h before electroporation to increase HDR efficiency. One hundred picomoles Cas9 protein and 130 pmol sgRNA were assembled into Cas9/sgRNA RNP complexes just before nucleofection and combined with 150 pmol HDR template in a final volume of 10 μL. Electroporation was performed in an Amaxa 96-well shuttle Nuleofector device (Lonza) using SF-cell line reagents (Lonza). Nocodazole-treated cells were resuspended to 10⁴ cells μL⁻¹ in SF solution immediately prior to electroporation. For each sample, 20 μL of cells was added to the 10 μL RNP/template mixture. Cells were immediately electroporated using the CM130 program and transferred to a 48-well plate with pre-warmed medium. Electroporated cells were cultured and expanded for 7–10 days prior to lentiviral transduction.

One day before lentiviral transduction, knocked-in cells well split and seeded in a 12-well plate at 6–8 × 10⁴ per well for four wells. The confluency should reach 70–80% on the day of transduction. The lentivirus titer of sfCherry2$_{1-10}$, sfCherry3C$_{1-10}$, or sfCherry3V$_{1-10}$ was quantified independently by Lenti-X GoStix Plus kit (Takara, Cat# 631280) immediately before infection. And the supernatant was diluted (around 1:5 dilution) with fresh medium to make sure the final virus concentrations are the same across groups. The experimental groups were treated with 1 ml diluted sfCherry2$_{1-10}$, sfCherry3C$_{1-10}$, or sfCherry3V$_{1-10}$ viral supernatant supplemented with 10 μg polybrene (MilliporeSigma), respectively. The negative control was treated with fresh medium supplemented with the same concentration of polybrene. Twenty-four after infection, the viral supernatant was swapped with fresh medium. After another 48–72 hours, the infected cells were harvested for flow cytometry analysis and cell sorting.

**Fluorescence microscopy.** In preparation of cell samples for the imaging purpose, to achieve better cell attachment, the eight-well glass bottom chamber (Thermo Fisher Scientific) was coated with Fibronectin (Sigma Aldrich) for 45 min, washed three times by PBS, and subjected to air-dry before seeding cells. To validate the performance of Spy-assisted complementation in enhancing the overall brightness of protein labeling (Fig. 2c, Supplementary Fig. 4), we transfected either HEK 293T cells or HeLa cells (0.8–1.5 × 10⁴ per well) grown on an eight-well chamber using FuGene HD according to the manufacturer's protocol (Promega). Total plasmid amount of 180 ng per well with the FP$_{11}$ to FP$_{1-10}$ ratio in 1:2 was used to achieve optimal expression and labeling. For SpyTag-sfCherry2$_{11}$ linked to the N-terminal of H2B, HEK 293T cells were fixed 48 h after transfection with 4% par-aformaldehyde and then imaged on a Nikon Ti-E inverted wide-field fluorescence microscope equipped with an LED light source (Excelitas X-Cite XLED1), a ×40 0.55 NA air objective (Nikon), a motorized stage (ASI), and an sCMOS camera (Tucsen). For sfCherry2$_{11}$-SpyTag linked to the C-terminal of TOMM20, 48 h after transfection, HeLa cells were imaged live on an inverted Nikon Ti-E microscope (UCSF Nikon Imaging Center), a Yokogawa CSU-W1 confocal scanner unit, a Plan Apo VC ×100 1.4 NA oil immersion objective, a stage incubator, an Andor Zyla

4.2 sCMOS camera, and MicroManager2.0 software. Live-cell imaging (Fig. 4b–g) of sorted successful knock-in HEK 293T cells (with either sfCherry3C$_{1–10}$ or sfCherry3V$_{1–10}$ infection) was acquired under the same condition.

Photoactivation of HEK 293T cells co-transfected with sfCherry2$_{11}$-H2B and PAsfCherry2$_{1–10}$ was acquired from the above-mentioned confocal microscope with the same setting. Photoactivation process was completed by 405 nm laser exposure. Mean fluorescence intensity (MFI) of each nucleus was quantified in ImageJ using an identical mask at three different time points: before activation, right after activation, and 10 min after activation (samples were kept still on the stage during this period). The absolute MFI of each nucleus at three time points was normalized to the intensity right after activation. Ten cells were analyzed to generate the bar graph in GraphPad Prism. Fluorescence recovery after photobleaching (FRAP) and photoactivation experiments (Supplementary Figs. 1 and 2) were performed on the same microscopy with a Vortran 473 nm laser (power = 9.0 mW) for photobleaching and a Rapp Optoelectronic UGA-40 photobleaching system (COM5). The frame interval of the movie is 2.5 s and the total length is 10 min. The photobleaching was conducted by scanning the 473 nm (50 mW) laser over the region of interest (marked by the yellow rectangular). A background image (taken at the same imaging condition without putting on any sample) was subtracted from the live-cell microscopy images using the ImageJ software. Analysis of fluorescence microscopy images were performed in ImageJ.

**Photobleaching analysis**. We measured the photobleaching properties of mCherry, split sfCherry3C, and split sfCherry3V in HEK 293T cells expressing these fluorescent proteins labeling the Lamin B structure, under the above-mentioned confocal microscope using a Plan Apo VC ×100 1.4 NA oil immersion objective. To trace the fluorescent intensities for live imaging of Lamins, we used Plot z-axis Profile function in ImageJ. For each photobleaching experiment, we calculated the MFI in a small ROI of lamin in one cell over the duration of the experiment (300 s), subtracted the background using a background image taken without samples under the same condition, and normalized the trajectory to the value of the first frame. Supplementary Fig. 7 reported the normalized photo-bleaching trajectories from five experiments for each construct (mCherry, split sfCherry3C, and split sfCherry3V). The half-life of each fluorescent proteins were calculated by fitting every trajectory to a single exponential decay model using GraphPad Prism.

**pH stability measurement**. To characterize the pH stability of split sfCherry3C and split sfCherry3V and compare them with full-length mCherry, we measured their fluorescent intensity in labeling the H2B structure of mammalian cells after fixation under physiological (pH = 7.4) and acidic (pH = 6 and 5) conditions. We used comercial phosphate-buffered saline (PBS, pH = 7.4, 1×, Gibco) as the phy-siological buffer. Acidic PBS (pH = 6) was prepared from PBS (pH = 7.4) and hydrochloric acid (Fisher Scientific); acetate buffer (pH = 5) was prepared from acetate sodium (Sigma Aldrich) and acetic acid (Acros Organics). pH value was measured by pH Meter (pH700, Oakton). HeLa cells were transfected with split sfCherry3C, split sfCherry3V, full-length mCherry in eight-well chamber (Thermo Fisher Scientific). After 45 h expression, cells were fixed with 4% paraformaldehyde for 10–15 min and kept in PBS (pH = 7.4) till imaging.

All images were acquired on a Nikon Ti Microscope, a Plan Fluor ×10 0.3 objective and 561 nm laser. For each sample, image of pH = 7.4 was firstly taken, then we gently changed buffer to acidic PBS (pH = 6) without disturbing the samples. After acquiring the image of pH = 6, the buffer was changed to acetate buffer (pH = 5) carefully, and the image of pH = 5 was taken. During the whole process of taking three images, the field of view was kept identical. Background images of microscope were acquired without samples. Images were analyzed in ImageJ. The background value was subtracted from every image by an Image calculator. For every sample, masks of cells were generated under pH = 7.4 and applied to all three pHs to get MFI of each cell under different pHs. The relative intensity ratio was normalized to pH = 7.4. Average of intensities of different samples were compared and plotted in GraphPad Prism. These intensities are directly proportional to the amount of chromophores in the deprotonated state. Therefore, we obtained the chromophore pKa value by fitting the intensity dependence on pH to a titration curve. The measured pKa for mCherry (4.8) is consistent with literature values, demonstrating the accuracy of our approach.

**Temperature stability measurement**. To characterize the temperature stability of split sfCherry2 and sfCherry3C, and compare them to the full-length sfCherry2, we measured their relative fluorescence intensity under physiological temperature (37 °C) and room temperature (25 °C) in living E. coli cultures. E. coli cells transformed with pETDuet-sfCherry2 (full length), pETDuet-sfCherry2$_{1–10}$-RBS-sfCherry2$_{11}$-SpyCatcher, or pETDuet-sfCherry3C$_{1–10}$-RBS-sfCherry2$_{11}$-SpyCatcher were first cultured overnight to saturation, then diluted back in 1 mM IPTG-containing medium to induce the protein expression. After 5 h protein induction under 37 °C or 25 °C, the fluorescence intensity (excitation at 561 nm, emission at 610 nm) and cell density (OD600) were acquired a Synergy™ H4 Hybrid Multi-Mode Microplate Reader at 37 or 25 °C. The fluorescence intensity of each con-struct was first normalized by the cell density and then normalized to the intensity at 37 °C. Data analysis and plotting were conducted using GradPad Prism.

**Generation of NLG-1 CLASP in C. elegans**. All C. elegans strains were maintained using standard protocols[35] and were raised on 60 mm Nematode Growth Media plates seeded with OP50 Escherichia coli at 20 °C. Wild-type strains were C. elegans variety Bristol, strain N2, and the mutant strain used for this study was clr-1(e1745) II. Transgenic strains include wyEx1982 (ref. [21]), wyEx1982; clr-1, iyEx368, iyEx368; clr-1, iyEx403 and iyEx403, clr-1. wyEx1982 contains the extra-chromosomal PHB-AVA NLG-1 GRASP marker ($_p$gpa-6::nlg-1::GFP$_{1–10}$ (60 ng μL$^{–1}$), $_p$flp-18::nlg-1:: GFP$_{11}$ (30 ng μL$^{–1}$), $_p$nlp-1::mCherry (10 ng μL$^{–1}$), $_p$flp-18::mCherry (5 ng μL$^{–1}$), and $_p$odr-1::DsRedII (20 ng μL$^{–1}$)). iyEx368 contains the extra-chromosomal PHB-AVA NLG-1 CLASP marker ($_p$gpa-6::nlg-1::sfCherry3C$_{1–10}$ (68 ng μL$^{–1}$), $_p$flp-18::nlg-1:: sfCherry2$_{11}$ (38.6 ng μL$^{–1}$), $_p$nlp-1::GFP (4.5 ng μL$^{–1}$), and $_p$odr-1::DsRedII (39 ng μL$^{–1}$)). iyEx403 contains the extra-chromosomal PHB-AVA NLG-1 CLASP marker ($_p$gpa-6::nlg-1::sfCherry3V$_{1–10}$ (73 ng μL$^{–1}$), $_p$flp-18::nlg-1::sfCherry2$_{11}$ (38.5 ng μL$^{–1}$)), and the AVA-VA/DA NLG-1 GRASP ($_p$unc-4::nlg-1::GFP$_{1–10}$ (20 ng μL$^{–1}$)[8], $_p$flp-18::nlg-1::GFP$_{11}$ (30 ng μL$^{–1}$))[8], and $_p$odr-1::DsRedII (39 ng μL$^{–1}$)).

A Zeiss Axio Imager.A1 compound fluorescent microscope was used to capture images of live C. elegans under ×63 magnification. Worms were paralyzed on 2% agarose pads using a 2:1 ratio of 0.3 M 2,3-butanedione monoxime (BDM) and 10 mM levamisole in M9 buffer. All micrographs taken of PHB-AVA NLG-1 GRASP and NLG-1 CLASP markers were of larval stage 4 animals. All data from micrographs were quantified using NIH ImageJ software. Intensity of PHB-AVA NLG-1 GRASP and PHB-AVA NLG-1 CLASP was measured as previously described[13,21]. Briefly, the intensity at each pixel within each synaptic puncta was measured using NIH ImageJ. To account for differences in background fluorescence, background intensity was estimated by calculating the minimum intensity value in a region immediately adjacent to the puncta. This minimum intensity value was then subtracted from the intensity for each pixel, and the sum of the adjusted values was calculated. For control, pictures of wild-type animals were also taken on the same day using the same settings.

**SDS behavior assay**. The SDS behavioral assay was used to test the function of the PHB circuit in Supplementary Fig. 10, and has been previously described[13,21]. Briefly, animals are placed on partially dried plates, and induced to move backwards by touching their nose with a hair-pick. A drop of M13 control buffer or 0.1% sodium dodecyl sulfate (SDS) diluted in M13 buffer is placed behind the animal, and quickly absorbs into the dry plate. The time the animal backs into the dry drop before stopping is then measured. It is important that the drop is dry so that it does not wick along the animal and activate sensory neurons outside the PHB circuit. Wild-type animals usually back up into M13 buffer for well over a second, but stop backing into 0.1% SDS in less than a second. Eighty wild-type and 80 marker-carrying animals were tested in this study. Forty animals from each genotype were tested with control M13 buffer, and 40 animals from each genotype were tested with 0.1% SDS in M13 buffer. The relative response index is calculated by dividing the response time to 0.1% SDS by the average response time to M13 buffer for the same genotype. The response index for marker-carrying animals was normalized to the response index for wild-type animals tested on the same day.

**Statistics and reproducibility**. For all characterizations of FP and split FP con-structs by flow cytometry, fluorescence fluctuation spectroscopy and microscopy, we have repeated the experiments with at least two biological replicates which have yielded consistent results.

**Reporting Summary**. Further information on research design is available in the Nature Research Reporting Summary linked to this article.

## Data availability

All data are available from the authors upon request (M.V. for data in Figs. 5 and 6 and Supplementary Fig. 10, J.D.M. for data in Fig. 1a, and B.H. for all other data). Fluorescent protein constructs are available from B.H. or AddGene. C. elegans lines are available upon request from M.V.

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

## Acknowledgements

We thank Dr. David Alexander Brown and Dr. Yina Wang for extensive discussion on data analysis, Dr. Bin Yang for help in fluorescence light microscopy, Alejandro Ramirez for preparing reagents and supplies, Dr. Kari Herrington for help in FRAP experiment, Dr. Noelle L'Etoile for advice on NLG-1 CLASP experiments, Dr. Xiaokun Shu for sharing the BioSpectrum Imaging System, Dr. Joseph DeRisi for generously sharing the nucleofector device, and Jordan Mitchell for contributing to the SDS behavior experiment. This work is supported by the National Institutes of Health R21EB022798 and R01GM124334 (to B.H.), UCSF Program for Breakthrough Biomedical Research (Byers Award in Basic Science to B.H.), the National Institutes of Health R01GM064589 (to J.K. and J.D.M.), the National Institutes of Health R01NS087544 and SC3GM089595 (to M. V.), T34GM008253 MARC fellowship (to N.A.), and the National Science Foundation 1355202 (to M.V.). B.H. is a Chan Zuckerberg Biohub investigator.

## Author contributions

S.F. and B.H. conceived and designed the research; S.F. performed the flow cytometry, random mutagenesis and screening, CRISPR knock-in, FACS, confocal microscopy, and FRAP experiments; S.Z. performed the pH stability and temperature stability experiments; J.K and J.D.M. performed single-molecule brightness measurement and analysis; S.F. and C.M. performed the molecular cloning and site-directed mutagenesis experiments; A.V., D.C.V., F.F., N.A. and M.V. performed the NLG-1 CLASP and NLG-1 GRASP experiments and analysis; S.F. and B.H. analyzed the data; S.F., M.V. and B.H. wrote the manuscript.
