## [Peer Review File · Communications Biology]

Reviewers' comments:

Reviewer #1 (Remarks to the Author):

This manuscript is presenting the substantial improvement of the split sfCherry system by 1) determinate reason of its low complementation efficiency and adopt the SpyCatcher-SpyTag system to aid the stability of the complementation, 2) screen variants of sfCherry with a brighter signal. The manuscript is well written in support of well-designed experiments to prove their hypothesis. Therefore, I would like to recommend this manuscript to be published in this journal. However, there are a couple of questions that the author should address to accept this manuscript.

1. The authors use SpyCatcher-SpyTag system to fuse to the N-terminal of sfCherry2(1-10) and sfCherry2(11). However, due to the function of a protein, some proteins of interest might not allow any modification at their N-terminal. Does the system the author introduce would work equally when they are fused to the C-terminal of the protein of interest? If the authors do not have any experimental support to answer this question, an assumption based on their excellent knowledge of the protein topology would be acceptable.

2. The authors used linkers in two different lengths between SpyCatcher and sfCherry2(1-10) and concluded that the shorter version is better without showing experimental data. However, Since the graphs in Suppl. Fig. 2 do not show the difference in the efficiency of these two linkers, they might need to explain more details on the reason for their pick. It would be of great help when readers would like to use this system for their protein of interest.

Reviewer #2 (Remarks to the Author):

This manuscript demonstrated two approaches to improve self-associating efficiency of split red fluorescent proteins (SAsFPs): (1) assistance through SpyTag/SpyCatcher interaction and (2) directed evolution, and also proposed a model to characterize the complementation process of FP fragments. Then the authors applied the evolved split sfCherry3 variants with improved overall fluorescence signal to visualize several subcellular structures or specific subsets of synapses in live animals. The methods are straightforward and the data in the manuscript are also clear. So, the newly evolved brighter split sfCherry3 variants will definitely aid endogenous protein or synapses labeling. However, I do have some questions and suggestions to the authors as following:

(1). In Fig.2, sfCherry211 was tagged to the N-terminus of POIs, is it possible to tag sfCherry211 to the C-terminus of POIs? Since functional residual sites (or protein domains) may locate in the N-terminus of some specific POIs. This is also true for some membrane proteins where fusion could only be made in one direction (e.g. C-terminus).

(2) For FP-based BiFC, in some cases, low temperature could largely influence the overall fluorescence signal especially to some red FP-based BiFC systems such as mCherry, mNeptune, could temperature influence the ensemble fluorescence signal of SpyTag/SpyCatcher assisted complementation of sfCherry2 or newly evolved sfCherry3 variants? (e.g. test and compare under 25°C and 37°C)

(3) The authors engineered and selected self-associating split sfCherry3 variants mainly based on improving self-associating efficiency and complemented overall fluorescence signal. So it is obvious and also as demonstrated that the overall complemented fluorescence signal was improved compared with previously established self-associating split sfCherry/sfCherry2 systems. However, except from

the "bright" side of FPs, other characteristics of red FP ("dark" side) are also important such as photostability (for long-term imaging), cytotoxicity, oligomerization tendency, pH stability and photochromism (for quantitative imaging). Although the authors have already shown some typical fluorescence images with correct localizations of endogenous protein labeling in HEK 293T cells using split sfCherry3 variants, it is better if the authors could provide more data or made discussion about those "dark" sides of their newly evolved red split sfCherry3 variants. I think those information are useful for extensive application of red split sfCherry3 (not only for endogenous structural labeling but also for functional bioimaging).

Reviewer #3 (Remarks to the Author):

In this work, Feng et al. developed sfCherry3V, a mCherry variant with improved spontaneous complementation when split into strands 1-10 and strand 11. They demonstrated brighter labeling of endogenous proteins labelled by strand 11 knock-in and created a red version of the synaptic labeling system GRASP which they named CLASP.

A robust spontaneously assembling RFP would be very useful, e.g. as a second channel for high-throughput protein labeling, or for a red GRASP system, exactly as the authors propose. The authors are to be commended for their hard work to improve technologies that would be broadly useful in biology. However the work described here falls a short in significance and rigor in important ways.

1. The FRAP experiment to support a two-step (reversible and irreversible) model of mCherry assembly is flawed. The authors use mNeonGreen FRAP and references to support their hypothesis that assembly of mCherry strands 1-10 with strand 11 is irreversible. Using mNeonGreen and FRAP is inappropriate because (1) mNeonGreen is not mCherry, and is not even close, as it is highly devolved and is GFP not a RFP, and (2) FRAP in any case could fail to show recovery if the strand 11 also is damaged in the course of FRAP, e.g. due to FALI.
2. References to support the two-step model of mCherry assembly are inappropriate or misinterpreted because (a) they are both about GFP, not mCherry, and (b) ref 1 does not present any data supporting irreversibility, and (b) ref 2 is for a different breakpoint (aa158).
3. Saying a two-step model would fit the trend of concentration dependence is not enough to prove it is a two-step assembly process. In fact, it could be that a simple two-component equilibrium model would be entirely sufficient to explain the finding that spontaneous reconstitution is not linearly related to expression level of the co-expressed IFP marker. The authors claim „On the other hand, a one-step, irreversible complementation would result in a direct proportional relationship between the complemented fluorescence signal and the fragment expression level.“ However the simple equilibrium relationship for reversible binding interactions between A and B is $[AB] = K_d[A][B]$. Clearly this is a non-linearly related to expression level of the IFP marker, since [A] and [B] both rise as IFP rises. To simplify, $[A] = [B]$ if they have similar stabilities, or more generally $[A] = c[B]$ if they have different stabilities, so then $[AB] = K_d * C * [A]^2$ proportional to $[A]^2$, or $[AB] = K_d[B]^2/C$ proportional to $[B]^2$. If models alone are to be used as evidence for mechanism, then this simpler alternative model should be explored and the authors can then demonstrate which one fits better.
4. The rigorous way to demonstrate a two-step assembly process would be to measure the actual rate of chromophore formation after complex formation and show that it is not dependent on concentration. This was done rigorously by Koker, Fernandez, and Pinaud for GFP1-10 + GFP11 in „Characterization of Split Fluorescent Protein Variants and Quantitative Analyses of Their Self-

Assembly Process, *Scientific Reports* 2018:5344.

5. Since affinity is the focus of the current study, actual dissociation constants should be measured in vitro and compared to split GFP affinity, as was also done by Koker et al above.

6. The CLASP work does not compare the new protein to the old one, so we don't know if it's really better in this application. CLASP should be done comparing sfCherry3Vm sfCherry3C, and sfCherry2 to know if the brightness improvement is real. Also could be done in the same animals with GRASP multiplexed, to see if CLASP is as sensitive as GRASP, across a variety of connections.

7. Increased affinity in GRASP creates a concern for increasing neuron-neuron adhesion and thus creating ectopic synapses. It is important to test a case where neurons touch but don't synapse to see if higher affinity causes spurious reactions. This was discussed in both the original GRASP and mGRASP papers. For example, it was found that cell cell contact alone was insufficient for a GRASP signal by looking at the mutant phenotypes, showing the GRASP signals changed as known synaptic sites changed with mutation (Feinberg et al *Neuron* 2008 57:353, Figures 4-5). For mGRASP this was done by EM as described in Kim et al, *Nature Methods* 2012 9:96, Supp Fig 6:

8. Regarding significance, the degree of enhancement of 3E not impressive, shows the problem is more unsolved than solved (less than halfway solved).

The last point (point 8) raises the question of when is an improvement worthy of reporting? I would suggest it is worthy of reporting if it makes a big enough difference to allow something to be done that couldn't have been done before. The work on endogenous cell labeling nicely shows a welcome improvement in sensitivity, but a similar comparison should be made for CLASP with sfmCherry3 vs its predecessors (point 6) and then the ability of the system to accurately label synapses should be confirmed (point 7).

Please see our responses to the comments in blue text.

Reviewer #1 (Remarks to the Author):

This manuscript is presenting the substantial improvement of the split sfCherry system by 1) determine reason of its low complementation efficiency and adopt the SpyCatcher-SpyTag system to aid the stability of the complementation, 2) screen variants of sfCherry with a brighter signal. The manuscript is well written in support of well-designed experiments to prove their hypothesis. Therefore, I would like to recommend this manuscript to be published in this journal. However, there are a couple of questions that the author should address to accept this manuscript.

We greatly appreciate the positive comments from the reviewer.

1. The authors use SpyCatcher-SpyTag system to fuse to the N-terminal of sfCherry2(1-10) and sfCherry2(11). However, due to the function of a protein, some proteins of interest might not allow any modification at their N-terminal. Does the system the author introduce would work equally when they are fused to the C-terminal of the protein of interest? If the authors do not have any experimental support to answer this question, an assumption based on their excellent knowledge of the protein topology would be acceptable.

In the revised manuscript, we have demonstrated that sfCherry2₁₁-SpyTag can also be fused to the C-terminus of protein of interest (TOMM20 in this case). We added this piece of data as the new Figure S4.

2. The authors used linkers in two different lengths between SpyCatcher and sfCherry2(1-10) and concluded that the shorter version is better without showing experimental data. However, Since the graphs in Suppl. Fig. 2 do not show the difference in the efficiency of these two linkers, they might need to explain more details on the reason for their pick. It would be of great help when readers would like to use this system for their protein of interest.

There was actually a subtle difference between the 15 a.a. linker and 6 a.a. linker in Figure S2 (now Figure S3) in that the data points in the 6 a.a. cases are slightly shifted upwards with respect to the reference line compared to the 15 a.a. cases. We do admit that this difference is very small. Therefore, in the main text, we added the following: "though the differences are subtle, indicating a flexibility in the linker choices and arrangement of the two domains".

Reviewer #2 (Remarks to the Author):

This manuscript demonstrated two approaches to improve self-associating efficiency of split red fluorescent proteins (SAsFPs): (1) assistance through SpyTag/SpyCatcher interaction and (2) directed evolution, and also proposed a model to characterize the complementation process of FP fragments. Then the authors applied the evolved split sfCherry3 variants with improved overall fluorescence signal to visualize several subcellular structures or specific subsets of synapses in live animals. The methods are straightforward and the data in the manuscript are also clear. So, the newly evolved brighter split sfCherry3 variants will definitely aid endogenous protein or synapses labeling. However, I do have some questions and suggestions to the authors as following:

Thanks a lot for the encouraging comments from the reviewer!

(1). In Fig.2, sfCherry211 was tagged to the N-terminus of POIs, is it possible to tag sfCherry211 to the C-terminus of POIs? Since functional residual sites (or protein domains) may locate in the N-terminus of some specific POIs. This is also true for some membrane proteins where fusion could only be made in one direction (e.g. C-terminus).

Reviewer 1 also brought up this point. In the revised manuscript, we have demonstrated the tagging of the C-terminus of TOMM20 using sfCherry2₁₁-SpyTag. This new piece of data is now added as the new Figure S4.

(2) For FP-based BiFC, in some cases, low temperature could largely influence the overall fluorescence signal especially to some red FP-based BiFC systems such as mCherry, mNeptune, could temperature influence the ensemble fluorescence signal of SpyTag/SpyCatcher assisted complementation of sfCherry2 or newly evolved sfCherry3 variants? (e.g. test and compare under 25 °C and 37 °C)

In the revised manuscript, we have compared the temperature dependence of fluorescence in full length and split versions of sfCherry2 as well as split sfCherry3C in *E. coli* cultures under either 37 °C or 25 °C (new Fig. S9). Although full length sfCherry2 demonstrated similar fluorescence under different temperatures, the split sfCherry2 had lower intensity under the 37 °C. However, split sfCherry3C was able to recover the fluorescence in the physiological temperature 37 °C, which makes it advantages for live-cell imaging applications.

(3) The authors engineered and selected self-associating split sfCherry3 variants mainly based on improving self-associating efficiency and complemented overall fluorescence signal. So it is obvious and also as demonstrated that the overall complemented fluorescence signal was improved compared with previously established self-associating split sfCherry/sfCherry2 systems. However, except from the “bright” side of FPs, other characteristics of red FP (“dark” side) are also important such as photostability (for long-term imaging), cytotoxicity, oligomerization tendency, pH stability and photochromism (for quantitative imaging). Although the authors have already shown some typical fluorescence images with correct localizations of endogenous protein labeling in HEK 293T cells using split sfCherry3 variants, it is better if the authors could provide more data or made discussion about those “dark” sides of their newly evolved red split sfCherry3 variants. I think those information are useful for extensive application of red split sfCherry3 (not only for endogenous structural labeling but also for functional bioimaging).

We fully agree with the reviewer on of the practical importance of such characterizations and greatly thank the reviewer for this suggestion. In the revised manuscript, we have compared the photobleaching rates of split sfCherry3C, sfCherry3V and full length mCherry, demonstrating that our new constructs have similar (sfCherry3V) or superior (sfCherry3C) photostability as mCherry (new Figure S7).

We have measured the pKa of split sfCherry3C and sfCherry3V (new Figure S8). Split sfCherry3C showed a relative low pKa (5.0), close to that of mCherry (pKa ~4.8) which is known to be acid-tolerant. On the other hand, split sfCherry3V presented a higher pKa (5.9), making it compatible with imaging at neutral pH but more sensitive to acidic environment such as in the lumen of lysosomes. This higher pKa explains the absence of lysosomal accumulation background using sfCherry3V labeling for ER proteins in Figure 4, contrasting images using sfCherry3C. As we and others have previously reported, long term labeling of endomembrane proteins using mCherry or its derivatives often lead to fluorescence accumulation in lysosomes because the mCherry scaffold resists lysosomal protease. sfCherry3V thus offers a solution to this problem.

We have examined the temperature dependence of chromophore maturation, showing that sfCherry2_{1-10/11} complements and matures less efficiently at 37 °C than 25 °C, whereas sfCherry3C is unaffected.

We have performed the standard OSER assay to characterize the oligomerization potential of the split sfCherry constructs. Unfortunately, because the FP₁₁ fragment of sfCherry3 is a short peptide, fusing it to the short signal peptide in the OSER assay leads to poor expression. Therefore, we were not able to characterize its oligomerization potential. Still, we note that full length sfCherry2 is likely has a low dimerization potential because its single-molecule brightness (Figure 1) is comparable to that of mCherry (our historical data) which has a high OSER monomer score.

We hope that these characterizations will provide useful information to guide the practical application of our newly engineering split sfCherry3C/V constructs.

Reviewer #3 (Remarks to the Author):

In this work, Feng et al. developed sfCherry3V, a mCherry variant with improved spontaneous complementation when split into strands 1-10 and strand 11. They demonstrated brighter labeling of endogenous proteins labelled by strand 11 knock-in and created a red version of the synaptic labeling system GRASP which they named CLASP.

A robust spontaneously assembling RFP would be very useful, e.g. as a second channel for high-throughput protein labeling, or for a red GRASP system, exactly as the authors propose. The authors are to be commended for their hard work to improve technologies that would be broadly useful in biology. However the work described here falls a short in significance and rigor in important ways.

1. The FRAP experiment to support a two-step (reversible and irreversible) model of mCherry assembly is flawed. The authors use mNeonGreen FRAP and references to support their hypothesis that assembly of mCherry strands 1-10 with strand 11 is irreversible. Using mNeonGreen and FRAP is inappropriate because (1) mNeonGreen is not mCherry, and is not even close, as it is highly devolved and is GFP not a RFP, and (2) FRAP in any case could fail to show recovery if the strand 11 also is damaged in the course of FRAP, e.g. due to FALI.

It is likely that our writing has generated some confusion or misunderstanding from the reviewer. In our original manuscript, we studied the complementation mechanism using split mNeonGreen2 as the model system, which led to the strategy to enhance the complementation signal by improving the (apparent) affinity between the two fragments. We then used this strategy to guide our further engineering on sfCherry2, which is in more urgent need for complementation brightness improvement. This strategy is unaffected by whether the complementation is ultimately reversible or irreversible, as the reviewer explains in point 3.

To avoid such confusions, we have now added new experiments to directly characterize the irreversibility of sfCherry2 complementation. We have performed the same FRAP experiment as we have previously done on split mNeonGreen2 (new Figure S1), showing no significant recovery of bleached sfCherry2_{1-10/11} fluorescence after 10 min. Moreover, in order to rule out the possibility that the lack of recovery is due to photobleaching-induced damage of the FP₁₁ fragment, we performed photoactivation experiment using sfCherry2₁₁-H2B and PAsfCherry2₁₋₁₀ (which adds the key point mutations in PAmCherry to sfCherry2₁₋₁₀, turning the complemented fluorescent protein into a photoactivatable one). In this experiment, 10 min after photoactivation, we saw no significant reduction of the H2B signal, indicating that there is minimal exchange of the photoactivated PAsfCherry2₁₋₁₀ fragment on this time scale. Both experiments suggest an irreversible overall complementation process for sfCherry2_{1-10/11}.

2. References to support the two-step model of mCherry assembly are inappropriate or misinterpreted because (a) they are both about GFP, not mCherry, and (b) ref 1 does not present any data supporting irreversibility, and (b) ref 2 is for a different breakpoint (aa158).

We must clarify that these two references by no means was used to support the two-step model. We mentioned them at the beginning of our Supplementary Note to state the general understanding for other, existing split fluorescent proteins. This was our motivation to study the reversibility of mNeonGreen2 complementation. We have modified the text of supplementary notes to avoid the misunderstanding that we are directly translating literature reports on other split fluorescent proteins onto the ones that we work on in this manuscript:

“The overall complementation process of commonly used split fluorescent proteins is known to be irreversible [1, 2], raising the possibility that the complementation of split mNG2_{1-10/11} and split sfCherry2_{1-10/11} is also irreversible. Here, we verified ...”

3. Saying a two-step model would fit the trend of concentration dependence is not enough to prove it is a two-step assembly process. In fact, it could be that a simple two-component equilibrium model would be entirely sufficient to explain the finding that spontaneous reconstitution is not linearly related to expression level of the co-expressed IFP marker. The authors claim, "On the other hand, a one-step, irreversible complementation would result in a direct proportional relationship between the complemented fluorescence signal and the fragment expression level.," However the simple equilibrium relationship for reversible binding interactions between A and B is $[AB] = K_d[A][B]$. Clearly this is a non-linearly related to expression level of the IFP marker, since [A] and [B] both rise as IFP rises. To simplify, $[A] = [B]$ if they have similar stabilities, or more generally $[A] = c[B]$ if they have different stabilities, so then $[AB] = K_d * C * [A]^2$ proportional to $[A]^2$, or

$[AB] = K_d[B]^2/C$ proportional to $[B]^2$. If models alone are to be used as evidence for mechanism, then this simpler alternative model should be explored and the authors can then demonstrate which one fits better.

We fully agree with the reviewer that a simple two-component equilibrium model could produce the same relationship between the complemented fluorescence and the fragment expression level. In fact, this simpler model was exactly our initial interpretation of the flow cytometry data. We had to modify it, though, after we confirmed the irreversibility of the overall complementation process. In the revised manuscript, we modified our text so that the one-step equilibrium model is also discussed. We also note that for the purpose of engineering a brighter fluorescent label (the ultimate goal of our manuscript), the difference between these two models does not matter.

4. The rigorous way to demonstrate a two-step assembly process would be to measure the actual rate of chromophore formation after complex formation and show that it is not dependent on concentration. This was done rigorously by Koker, Fernandez, and Pinaud for GFP1-10 + GFP11 in , Characterization of Split Fluorescent Protein Variants and Quantitative Analyses of Their Self-Assembly Process, Scientific Reports 2018:5344.

Indeed we have tried to purify sfCherry2₁₋₁₀ fragment for *in vitro* analysis exactly in the same way as in the paper mentioned by the reviewer. However, sfCherry2₁₋₁₀ is much less soluble than GFP₁₋₁₀ (which we have previously purified), making the yield too low to practically perform biochemical characterization of the complementation process. As much as we would love to have these detailed kinetics parameters, we do not think it is practical within a reasonable time frame to revise this manuscript.

We also note that the paper mentioned by the reviewer has the specific data showing that GFP_{1-10/11} complementation is irreversible. Therefore, we added it to our references.

5. Since affinity is the focus of the current study, actual dissociation constants should be measured *in vitro* and compared to split GFP affinity, as was also done by Koker et al above.

In fact, Koker et al. did not measure dissociation constants because it cannot be done *in vitro* for the irreversible process of split GFP complementation. Instead, they measured just the binding rate constant (k_{on}) and maturation rate constant (k_{mat}). Only in cells with this process reaching a steady state, we can define an effective dissociation constant, K_D' , as discussed in our Supplementary Note. This effective dissociation constant can be read out from the flow cytometry scatter plot as the x axis position when the point distribution changes from a slope of 1 to slope of 2. We can only perform a crude estimation, though because the expression level range from our transient expression system is limited.

For GFP_{1-10/11}, the flow cytometry point distribution actually has a slight downward bend away from the slope-1 trend line at the low expression end (Fig. 1B). The position of this bend gives an approximate estimation of the K_D' to be $\sim 10^3$ (in the units of mFIP fluorescence. Same unit below). For sfCherry2_{1-10/11}, the full-length (slope = 1) and split (slope = 2) trend lines cross at approximately $K_D' = 3 \times 10^5$ (Fig. 1D). For sfCherry3V_{1-10/11}, there is a slight downward bend at the high concentration end (Fig. 3E), giving an approximate K_D' of $\sim 3 \times 10^4$. Taken together, the effective dissociation constant for sfCherry3V_{1-10/11} is approximately 30 times of that of GFP_{1-10/11}, compared to ~ 300 times in the case of sfCherry2_{1-10/11}. This ~ 10 fold improvement is consistent with our direct comparison (8.2 fold) measured in Figure 3E. We have added these discussions to the Supplementary Notes.

6. The CLASP work does not compare the new protein to the old one, so we don't know if it's really better in this application. CLASP should be done comparing sfCherry3Vm sfCherry3C, and sfCherry2 to know if the brightness improvement is real. Also could be done in the same animals with GRASP multiplexed, to see if CLASP is as sensitive as GRASP, across a variety of connections.

We must clarify that the significance in the CLASP work is in that we are the first to develop a synapse detection method in the red color channel, thus creating a brand new color channel to enable simultaneous visualization of synapses among multiple neurons. As suggested by the reviewer, we have added multiplexed imaging with GRASP to demonstrate this new and important capability. Please also see response to the very last comment about the significance of this capability.

Because there has never been an “old one” for CLASP, we do not think we need to (or should) compare the CLASP signal using sfCherry3V versus the previously developed sfCherry2, which has never been use in this type of applications.

7. Increased affinity in GRASP creates a concern for increasing neuron-neuron adhesion and thus creating ectopic synapses. It is important to test a case where neurons touch but don't synapse to see if higher affinity causes spurious reactions. This was discussed in both the original GRASP and mGRASP papers. For example, it was found that cell cell contact alone was insufficient for a GRASP signal by looking at the mutant phenotypes, showing the GRASP signals changed as known synaptic sites changed with mutation (Feinberg et al Neuron 2008 57:353, Figures 4-5). For mGRASP this was done by EM as described in Kim et al, Nature Methods 2012 9:96, Supp Fig 6:

Inducing additional synapses between PHB and AVA potentiates the response to the PHB-sensed chemical sodium dodecyl sulfate (SDS), as observed in *unc-6/Netrin* overexpressing animals (Park et

al., 2011) and *clr-1/RPTP* overexpressing animals (Varshney et al., 2018). In either genotype, animals respond to SDS more rapidly, so that the relative response index is significantly lower than that of wild-type animals. We previously determined that PHB-AVA NLG-1 GRASP did not induce additional synaptogenesis by testing animals carrying the marker for their response to SDS, and found that the response in PHB-AVA NLG-1 GRASP-carrying animals did not differ significantly from that of wild type animals (Park et al., 2011). To address this concern about PHB-AVA NLG-1 CLASP, we similarly performed SDS behavioral analysis on animals carrying this marker, as well as AVA-VA/DA NLG-1 GRASP marker. The response of PHB-AVA NLG-1 CLASP-carrying animals to SDS was not significantly different from that of non-marker carrying wild type animals, similarly indicating that additional synaptogenesis is not induced by NLG-1 CLASP (Fig. S10). In addition, we demonstrate the PHB-AVA NLG-1 CLASP puncta are similar in spatial distribution to NLG-1 GRASP puncta (Figure 5), and as with NLG-1 GRASP puncta, are severely reduced in *clr-1/RPTP* loss-of-function mutants (Figure 5), also consistent with these puncta representing endogenous synapses.

8. Regarding significance, the degree of enhancement of 3E not impressive, shows the problem is more unsolved than solved (less than halfway solved).

It is very likely that our description of Figure 3E results understated the performance enhancement by split sfCherry3C and split sfCherry3V. Under non-saturated-binding conditions, which is relevant for most practical applications, the change in the apparent affinity is directly proportional to the change in complementation efficiency and complemented signal (see Supplementary Note). Therefore, Figure 3E actually shows a brightness improvement of 2.5 to 8.2 fold, which is highly significant for fluorescent labeling. This is exactly the reason why we have obtained a dramatic improvement of endogenous cell sorting sensitivity. Previously, we focused the description on the change in effective affinity instead of signal brightness for the purpose of being accurate, because the practical fold of improvement when labeling endogenous proteins may vary because of the difference in protein turnover rates. We have revised the text to clarify the connection between the change in apparent affinity and overall labeling brightness to avoid potential misreading of this piece of data.

In terms of solving the problem of incomplete complementation, we have presented the solution using tandem sfCherry2₁₁-SpyTag. Moreover, in Figure 3E, sfCherry3V has already shown the sign of saturated complementation at high expression levels, meaning that it has nearly reached the labeling signal of the full length counterpart.

The last point (point 8) raises the question of when is an improvement worthy of reporting? I would suggest it is worthy of reporting if it makes a big enough difference to allow something to be done that couldn't have been done before. The work on endogenous cell labeling nicely shows a welcome improvement in sensitivity, but a similar comparison should be made for CLASP with sfCherry3 vs its predecessors (point 6) and then the ability of the system to accurately label synapses should be confirmed (point 7).

As we have explained in addressing point 8, the complementation signal improvement from sfCherry3C/V is highly significant. The improvement in sorting efficiency for endogenous protein labeling is the direct result of this large signal enhancement. We tested four targets with reasonably high level of expression so that sfCherry2_{1-10/11} complementation can still produce positive signals for quantitative comparison. The majority of the human proteome is actually expressed at a lower level than the lowest expressing one of the four targets, LMNA, which is barely detectable with sfCherry2_{1-10/11}. Therefore, it is clear that our improved signal will enable red-colored endogenous tagging of a large number of protein targets that cannot be done previously.

The development of NLG-1 CLASP is also new and significant, and this application is far better than having only NLG-1 GRASP at our disposal, or yellow and cyan variants. This advance greatly facilitates the labeling of different groups of specific synapses within the same animal, which will improve our ability to understand the function of genes in synaptic formation, regulation, and removal in development, as well as in contexts such as learning and memory. In fact, we take this even further, demonstrating that NLG-1 CLASP, in combination with NLG-1 GRASP, allows us to simultaneously visualize synapses from the same neuron onto different neuronal partners. This is a critical advance in this transparent model organism, as it will allow us to better understand how different groups of synapses are affected by single genes within individual animals (Figure 6). This marker will be of use to scientists studying nervous system development, homeostasis, aging, learning and memory, and diseases in which any of these processes are affected.

REVIEWERS' COMMENTS:

Reviewer #2 (Remarks to the Author):

In the revised manuscript "Bright split red fluorescent proteins for the visualization of endogenous proteins and synapses", the authors have conducted many new experiments and analyses. These new data have addressed all my concerns in the previous round of reviewing, and I would like to recommend acceptance.

Reviewer #3 (Remarks to the Author):

The authors' answers to my concerns, while not ideal, are adequate.

By the way, they agreed with my comment that the sentences "On the other hand, a one-step, irreversible complementation would result in a direct proportional relationship between the complemented fluorescence signal and the fragment expression level. In this case, the FACS data points in Figure 1C (left) should follow a trend line with a slope of 1, whereas in reality, they fall closer to a trend line with a slope of 2." are completely incorrect. These sentences should be removed before publication.

Other than that, I am in support of publication.

REVIEWERS' COMMENTS:

Reviewer #2 (Remarks to the Author):

In the revised manuscript “Bright split red fluorescent proteins for the visualization of endogenous proteins and synapses”, the authors have conducted many new experiments and analyses. These new data have addressed all my concerns in the previous round of reviewing, and I would like to recommend acceptance.

We thank the reviewer for the support!

Reviewer #3 (Remarks to the Author):

The authors' answers to my concerns, while not ideal, are adequate.

We thank the reviewer for recognizing our responses in the revision!

By the way, they agreed with my comment that the sentences "On the other hand, a one-step, irreversible complementation would result in a direct proportional relationship between the complemented fluorescence signal and the fragment expression level. In this case, the FACS data points in Figure 1C (left) should follow a trend line with a slope of 1, whereas in reality, they fall closer to a trend line with a slope of 2." are completely incorrect. These sentences should be removed before publication.

Regarding this sentence in the the Supplementary Note, I am afraid that the reviewer has misread “irreversible” in our writing as “reversible” because of the phase “one-step”. As we have explained previously, we agreed with the reviewer that a *one-step, reversible* complementation process would produce a trend line with a slope of 2. The reviewer also agreed with us that a *two-step, irreversible* complementation process produces the same result, though it is a more complicated mechanism. However, this sentence was mentioning a *one-step, irreversible* complementation process, which is distinct from both cases above. To avoid such confusion, we have now changed the phrase in question into “*directly irreversible* complementation”.

Other than that, I am in support of publication.